# Adaptive Online Estimation of Piecewise Polynomial Trends

**Dheeraj Baby**
Department of Computer Science
UC Santa Barbara
dheeraj@ucsb.edu

**Yu-Xiang Wang**
Department of Computer Science
UC Santa Barbara
yuxiangw@cs.ucsb.edu

## Abstract

We consider the framework of non-stationary stochastic optimization [Besbes et al., 2015] with squared error losses and noisy gradient feedback where the dynamic regret of an online learner against a time varying comparator sequence is studied. Motivated from the theory of non-parametric regression, we introduce a *new variational constraint* that enforces the comparator sequence to belong to a discrete $k^{th}$ order Total Variation ball of radius $C_n$. This variational constraint models comparators that have piecewise polynomial structure which has many relevant practical applications [Tibshirani, 2014]. By establishing connections to the theory of wavelet based non-parametric regression, we design a *polynomial time* algorithm that achieves the nearly *optimal dynamic regret* of $\tilde{O}(n^{\frac{1}{2k+3}} C_n^{\frac{2}{2k+3}})$. The proposed policy is *adaptive to the unknown radius $C_n$*. Further, we show that the same policy is minimax optimal for several other non-parametric families of interest.

## 1   Introduction

In time series analysis, estimating and removing the trend are often the first steps taken to make the sequence "stationary". The non-parametric assumption that the underlying trend is a piecewise polynomial or a spline [de Boor, 1978], is one of the most popular choices, especially when we do not know where the "change points" are and how many of them are appropriate. The higher order Total Variation (see Assumption A3) of the trend can capture in some sense both the sparsity and intensity of changes in underlying dynamics. A non-parametric regression method that penalizes this quantity — trend filtering [Tibshirani, 2014] — enjoys a superior *local adaptivity* over traditional methods such as the Hodrick-Prescott Filter [Hodrick and Prescott, 1997]. However, Trend Filtering is an *offline* algorithm which limits its applicability for the inherently *online* time series forecasting problem. In this paper, we are interested in designing an online forecasting strategy that can essentially match the performance of the offline methods for trend estimation, hence allowing us to apply time series models forecasting on-the-fly. In particular, our problem setup (see Figure 1) and algorithm are applicable to all *online variants* of trend filtering problem such as predicting stock prices, server payloads, sales etc.

Let's describe the notations that will be used throughout the paper. All vectors and matrices will be written in bold face letters. For a vector $\boldsymbol{x} \in \mathbb{R}^m$, $\boldsymbol{x}[i]$ or $\boldsymbol{x}_i$ denotes its value at the $i^{th}$ coordinate. $\boldsymbol{x}[a:b]$ or $\boldsymbol{x}_{a:b}$ is the vector $[\boldsymbol{x}[a], \ldots, \boldsymbol{x}[b]]$. $\|\cdot\|_p$ denotes finite dimensional $L_p$ norms. $\|\boldsymbol{x}\|_0$ is the number of non-zero coordinates of a vector $\boldsymbol{x}$. $[n]$ represents the set $\{1, \ldots, n\}$. $\boldsymbol{D}^i \in \mathbb{R}^{(n-i) \times n}$ denotes the discrete difference operator of order $i$ defined as in [Tibshirani, 2014] and reproduced

below.

$$\boldsymbol{D}^1 = \begin{bmatrix} -1 & 1 & 0 & \ldots & 0 & 0 \\ 0 & -1 & 1 & \ldots & 0 & 0 \\ \vdots & & & & & \\ 0 & 0 & 0 & \ldots & -1 & 1 \end{bmatrix} \in \mathbb{R}^{(n-1)\times n},$$

and $\boldsymbol{D}^i = \tilde{\boldsymbol{D}}^1 \cdot \boldsymbol{D}^{i-1} \ \forall i \geq 2$ where $\tilde{\boldsymbol{D}}^1$ is the $(n-i) \times (n-i+1)$ truncation of $\boldsymbol{D}^1$.

The theme of this paper builds on the non-parametric online forecasting model developed in [Baby and Wang, 2019]. We consider a sequential $n$ step interaction process between an agent and an adversary as shown in Figure 1.

---

1. Fix a time horizon $n$.
2. Agent declares a forecasting strategy $\mathcal{S}$
3. Adversary chooses a sequence $\boldsymbol{\theta}_{1:n}$
4. For $t = 1, \ldots, n$:
   (a) Agent outputs a prediction $\mathcal{S}(t)$.
   (b) Adversary reveals $y_t = \boldsymbol{\theta}_{1:n}[t] + \epsilon_t$
5. After $n$ steps, agent suffers a cumulative loss $\sum_{i=1}^{n} \left( \mathcal{S}(i) - \boldsymbol{\theta}_{1:n}[i] \right)^2$.

---

Figure 1: Interaction protocol

A forecasting strategy $\mathcal{S}$ is defined as an algorithm that outputs a prediction $\mathcal{S}(t)$ at time $t$ only based on the information available after the completion of time $t-1$. Random variables $\epsilon_t$ for $t \in [n]$ are independent and subgaussian with parameter $\sigma^2$. This sequential game can be regarded as an online version of the non-parametric regression setup well studied in statistics community.

In this paper, we consider the problem of forecasting sequences that obey $n^k \|D^{k+1}\boldsymbol{\theta}_{1:n}\|_1 \leq C_n$, $k \geq 0$ and $\|\boldsymbol{\theta}_{1:n}\|_\infty \leq B$. The constraint $n^k \|D^{k+1}\boldsymbol{\theta}_{1:n}\|_1 \leq C_n$ has been widely used in the rich literature of non-parametric regression. For example, the offline problem of estimating sequences obeying such higher order difference constraint from noisy labels under squared error loss is studied in [Mammen and van de Geer, 1997, Donoho et al., 1998, Tibshirani, 2014, Wang et al., 2016, Sadhanala et al., 2016, Guntuboyina et al., 2017] to cite a few. We aim to design forecasters whose predictions are only based on past history and still perform as good as a batch estimator that sees the entire observations ahead of time.

**Scaling of $n^k$.** The family $\{\boldsymbol{\theta}_{1:n} \mid n^k \|D^{k+1}\boldsymbol{\theta}_{1:n}\|_1 \leq C_n\}$ may appear to be alarmingly restrictive for a constant $C_n$ due to the scaling factor $n^k$, but let us argue why this is actually a natural construct. The continuous $TV^k$ distance of a function $f : [0,1] \to \mathbb{R}$ is defined as $\int_0^1 |f^{(k+1)}(x)| dx$, where $f^{(k+1)}$ is the $(k+1)^{th}$ order (weak) derivative. A sequence can be obtained by sampling the function at $x_i = i/n$, $i \in [n]$. Discretizing the integral yields the $TV^k$ distance of this sequence to be $n^k \|D^{k+1}\boldsymbol{\theta}_{1:n}\|_1$. Thus, the $n^k \|D^{k+1}\boldsymbol{\theta}_{1:n}\|_1$ term can be interpreted as the discrete approximation to continuous higher order TV distance of a function. See Figure 2 for an illustration for the case $k = 1$.

**Non-stationary Stochastic Optimization.** The setting above can also be viewed under the framework of non-stationary stochastic optimization as studied in [Besbes et al., 2015, Chen et al., 2018b] with squared error loss and noisy gradient feedback. At each time step, the adversary chooses a loss function $f_t(x) = (x - \boldsymbol{\theta}_t)^2$. Since $\nabla f_t(x) = 2(x - \boldsymbol{\theta}_t)$, the feedback $\tilde{\nabla} f_t(x) = 2(x - y_t)$ constitutes an unbiased estimate of the gradient $\nabla f_t(x)$. [Besbes et al., 2015, Chen et al., 2018b] quantifies the performance of a forecasting strategy $\mathcal{S}$ in terms of dynamic regret as follows.

$$R_{dynamic}(\mathcal{S}, \boldsymbol{\theta}_{1:n}) := \mathbb{E}\left[\sum_{t=1}^{n} f_t\left(\mathcal{S}(t)\right)\right] - \sum_{t=1}^{n} \inf_{x_t} f_t(x_t), = \mathbb{E}\left[\sum_{t=1}^{n} \left(\mathcal{S}(t) - \boldsymbol{\theta}_{1:n}[t]\right)^2\right], \quad (1)$$

where the last equality follows from the fact that when $f_t(x) = (x - \boldsymbol{\theta}_{1:n}[t])^2$, $\inf_x (x - \boldsymbol{\theta}_{1:n}[t])^2 = 0$. The expectation above is taken over the randomness in the noisy gradient feedback and that of the agent's forecasting strategy. It is impossible to achieve sublinear dynamic regret against arbitrary

ground truth sequences. However if the sequence of minimizers of loss functions $f_t(x) = (x - \theta_t)^2$ obey a path variational constraint, then we can parameterize the dynamic regret as a function of the path length, which could be sublinear when the path-length is sublinear. Typical variational constraints considered in the existing work includes $\sum_t |\boldsymbol{\theta}_t - \boldsymbol{\theta}_{t-1}|$, $\sum_t |\boldsymbol{\theta}_t - \boldsymbol{\theta}_{t-1}|^2$, $(\sum_t \|f_t - f_{t-1}\|_p^q)^{1/q}$ [see Baby and Wang, 2019, for a review]. These are all useful in their respective contexts, but do not capture higher order smoothness.

The purpose of this work is to connect ideas from batch non-parametric regression to the framework of online stochastic optimization and define a *natural family of higher order variational functionals* of the form $\|D^{k+1}\boldsymbol{\theta}_{1:n}\|_1$ to track a comparator sequence with piecewise polynomial structure. To the best of our knowledge such higher order path variationals for $k \geq 1$ are vastly unexplored in the domain of non-stationary stochastic optimization. In this work, we take the first steps in introducing such variational constraints to online non-stationary stochastic optimization and exploiting them to get sub-linear dynamic regret.

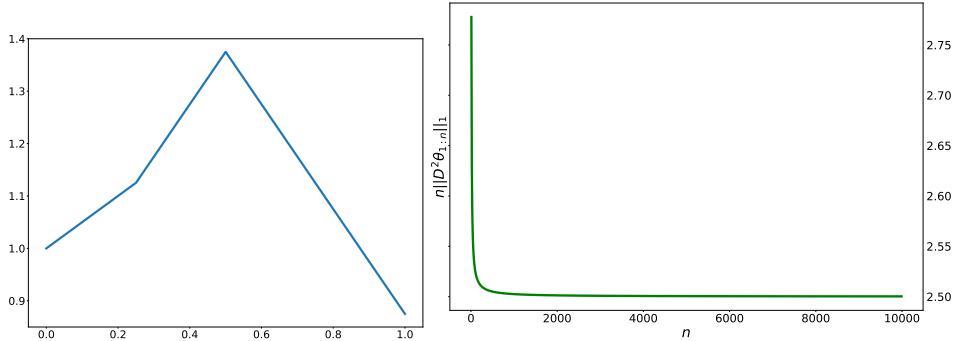

Figure 2: *A $TV^1$ bounded comparator sequence $\boldsymbol{\theta}_{1:n}$ can be obtained by sampling the continuous piecewise linear function on the left at points $i/n$, $i \in [n]$. On the right, we plot the $TV^1$ distance (which is equal to $n\|D^2\boldsymbol{\theta}_{1:n}\|_1$ by definition) of the generated sequence for various sequence lengths $n$. As $n$ increases the discrete $TV^1$ distance converges to a constant value given by the continous $TV^1$ distance of the function on left panel.*

## 2 Summary of results

In this section, we summarize the assumptions and main results of the paper.

**Assumptions.** We start by listing the assumptions made and provide justifications for them.

(A1) The time horizon is known to be $n$.

(A2) The parameter $\sigma^2$ of subgaussian noise in the observations is a known fixed positive constant.

(A3) The ground truth denoted by $\boldsymbol{\theta}_{1:n}$ has its $k^{th}$ order total variation bounded by some positive $C_n$, i.e., we consider ground truth sequences that belongs to the class

$$\text{TV}^k(C_n) := \{\boldsymbol{\theta}_{1:n} \in \mathbb{R}^n : n^k \|D^{k+1}\boldsymbol{\theta}_{1:n}\|_1 \leq C_n\}$$

We refer to $n^k \|D^{k+1}\boldsymbol{\theta}_{1:n}\|_1$ as $TV^k$ distance of the sequence $\boldsymbol{\theta}_{1:n}$. To avoid trivial cases, we assume $C_n = \Omega(1)$.

(A4) The TV order $k$ is a known *fixed* positive constant.

(A5) $\|\boldsymbol{\theta}_{1:n}\|_\infty \leq B$ for a known *fixed* positive constant $B$.

Though we require the time horizon to be known in advance in assumption (A1), this can be easily lifted using standard doubling trick arguments. The knowledge of time horizon helps us to present the policy in a most transparent way. If standard deviation of sub-gaussian noise is unknown, contrary to assumption (A2), then it can be robustly estimated by a Median Absolute Deviation estimator using first few observations, see for eg. Johnstone [2017]. This is indeed facilitated by the sparsity of wavelet coefficients of $TV^k$ bounded sequences. Assumption (A3) characterizes the ground truth

sequences whose forecasting is the main theme of this paper. The $\mathrm{TV}^k(C_n)$ class features a rich family of sequences that can potentially exhibit spatially non-homogeneous smoothness. For example it can capture sequences that are piecewise polynomials of degree at most $k$. This poses a challenge to design forecasters that are *locally adaptive* and can efficiently detect and make predictions under the presence of the non-homogeneous trends. Though knowledge of the TV order $k$ is required in assumption (A4), most of the practical interest is often limited to the lower orders $k = 0, 1, 2, 3$, see for eg. [Kim et al., 2009, Tibshirani, 2014] and we present (in Appendix D) a meta-policy based on exponential weighted averages [Cesa-Bianchi and Lugosi, 2006] to adapt to these lower orders. Finally assumption (A5) is standard in the online learning literature.

**Our contributions.** We summarize our main results below.

- When the revealed labels are noisy realizations of sequences that belong to $TV^k(C_n)$ we propose a *polynomial time* policy called `Ada-VAW` (**Ada**ptive **V**ovk **A**zoury **W**armuth forecaster) that achieves the nearly *minimax optimal* rate of $\tilde{O}\left(n^{\frac{1}{2k+3}} C_n^{\frac{2}{2k+3}}\right)$ for $R_{dynamic}$ with high probability. The proposed policy *optimally adapts to the unknown radius $C_n$*.

- We show that the proposed policy achieves optimal $R_{dynamic}$ when revealed labels are noisy realizations of sequences residing in higher order discrete Holder and discrete Sobolev classes.

- When the revealed labels are noisy realizations of sequences that obey $\|D^k\boldsymbol{\theta}_{1:n}\|_0 \le J_n, \|\boldsymbol{\theta}_{1:n}\|_\infty \le B$, we show that the same policy achieves the minimax optimal $\tilde{O}(J_n)$ rate for for $R_{dynamic}$ with high probability. The policy *optimally adapts to unknown $J_n$*.

**Notes on key novelties.** It is known that the VAW forecaster is an optimal algorithm for online polynomial regression with squared error losses [Cesa-Bianchi and Lugosi, 2006]. With the side information of change points where the underlying ground truth switches from one polynomial to another, we can run a VAW forecaster on each of the stable polynomial sections to control the cumulative squared error of the policy. We use the machinery of wavelets to mimic an oracle that can provide side information of the change points. For detecting change points, a restart rule is formulated by exploiting connections between wavelet coefficients and locally adaptive regression splines. This is a *more general* strategy than that used in [Baby and Wang, 2019]. To the best of our knowledge, this is the *first* time an interplay between VAW forecaster and theory of wavelets along with its adaptive minimaxity [Donoho et al., 1998] has been used in the literature.

Wavelet computations require the length of underlying data whose wavelet transform needs to be computed has to be a power of 2. In practice this is achieved by a padding strategy in cases where original data length is not a power of 2. We show that most commonly used padding strategies – eg. zero padding as in [Baby and Wang, 2019] – are not useful for the current problem and propose a novel *packing strategy* that alleviates the need to pad. This will be useful to many applications that use wavelets which can be well beyond the scope of the current paper.

Our proof techniques for bounding regret use properties of the CDJV wavelet construction [Cohen et al., 1993]. To the best of our knowledge, this is the *first* time we witness the ideas from a general CDJV construction scheme implying useful results in an online learning paradigm. Optimally controlling the bias of VAW demands to carefully bound the $\ell_2$ norm of coefficients computed by polynomial regression. This is done by using ideas from number theory and symbolic determinant evaluation of polynomial matrices. This could be of independent interest in both offline and online polynomial regression.

## 3 Related Work

In this section, we briefly discuss the related work. A discussion on preliminaries and a detailed exposition of related literature is deferred to Appendix A and B respectively. Throughout this paper when we refer as $\tilde{O}(n^{\frac{1}{2k+3}})$ as optimal regret we assume that $C_n = n^k\|D^{k+1}\boldsymbol{\theta}_{1:n}\|_1$ is $O(1)$.

**Non-parametric Regression** As noted in Section 1, the problem setup we consider can be regarded as an online version of the batch non-parametric regression framework. It has been established (see for eg, [Mammen and van de Geer, 1997, Donoho et al., 1998, Tibshirani, 2014] that minimax rate for estimating sequences with bounded $TV^k$ distance under squared error loss scales as

$n^{\frac{1}{2k+3}}(n^k\|D^{k+1}\boldsymbol{\theta}_{1:n}\|_1)^{\frac{2}{2k+3}}$ modulo logarithmic factors of $n$. In this work, we aim to achieve the same rate for minimax dynamic regret in online setting.

**Non-stationary Stochastic Optimization** Our forecasting framework can be considered as a special case of non-stationary stochastic optimization setting studied in [Besbes et al., 2015, Chen et al., 2018b]. It can be shown that their proposed algorithm namely, restarting Online Gradient Descend (OGD) yields a suboptimal dynamic regret of $O\left(n^{1/2}(\|D\boldsymbol{\theta}_{1:n}\|_1)^{1/2}\right)$ for our problem. However, it should be noted that their algorithm works with general strongly convex and convex losses. A summary of dynamic regret of various algorithms are presented in Table 1. The rationale behind how to translate existing regret bounds to our setting is elaborated in Appendix B.

Table 1: *Regret bounds for sequences that satisfy $n^k\|D^{k+1}\boldsymbol{\theta}_{1:n}\|_1\leq C_n$ with $\boldsymbol{\theta}[1:k+1]=0$, $\|\boldsymbol{\theta}_{1:n}\|_\infty\leq B$ and $k\geq 1$. The proposed policy doesn't require the knowledge of $C_n$ apriori while still attains the optimal dynamic regret modulo log factors. The bound for* `Ada-VAW` *holds even without the constraint on initial sequence values.*

| Policy | Dynamic Regret | Known $C_n$? | Lower bound |
|---|---|---|---|
| Moving Averages, Restarting OGD [Besbes et al., 2015] | $\tilde{O}(\sqrt{nC_n})$ | Yes | |
| OGD [Zinkevich, 2003] | $\tilde{O}(\sqrt{nC_n})$ | Yes | $\Omega\left(n^{1/(2k+3)}C_n^{2/(2k+3)}\right)$ |
| Ader [Zhang et al., 2018a] | $\tilde{O}(\sqrt{nC_n})$ | No | |
| Arrows [Baby and Wang, 2019] | $\tilde{O}\left(n^{1/3}C_n^{2/3}\right)$ | No | |
| `Ada-VAW` (**This paper**) | $\tilde{O}\left(n^{1/(2k+3)}C_n^{2/(2k+3)}\right)$ | No | |

**Prediction of Bounded Variation sequences** Our problem setup is identical to that of [Baby and Wang, 2019] except for the fact that they consider forecasting sequences whose zeroth order Total Variation is bounded. Our work can be considered as a generalization to any TV order $k$. Their algorithm gives a suboptimal regret of $O(n^{1/3}\|D\boldsymbol{\theta}_{1:n}\|_1^{2/3})$ for $k\geq 1$.

**Competitive Online Non-parametric Regression** [Rakhlin and Sridharan, 2014] considers an online learning framework with squared error losses where the learner competes against the best function in a non-parametric function class. Their results imply via a non-constructive argument, the existence of an algorithm that achieves the regret of $\tilde{O}(n^{\frac{1}{2k+3}})$ for our problem.

# 4 Main results

We present below the main results of the paper. All proofs are deferred to the appendix.

## 4.1 Limitations of linear forecasters

We exhibit a lower-bound on the dynamic regret that is implied by [Donoho et al., 1998] in batch regression setting.

**Proposition 1** (Minimax Regret)**.** *Let $y_t = \boldsymbol{\theta}_{1:n}[t] + \epsilon_t$ for $t = 1, \ldots, n$ where $\theta_{1:n} \in TV^{(k)}(C_n)$, $|\boldsymbol{\theta}_{1:n}[t]|\leq B$ and $\epsilon_t$ are iid $\sigma^2$ subgaussian random variables. Let $\mathcal{A}_F$ be the class of all forecasting strategies whose prediction at time $t$ only depends on $y_1, \ldots, y_{t-1}$. Let $\boldsymbol{s_t}$ denote the prediction at time $t$ for a strategy $\boldsymbol{s} \in \mathcal{A}_F$. Then,*

$$\inf_{\boldsymbol{s}\in\mathcal{A}_F} \sup_{\boldsymbol{\theta}_{1:n}\in TV^{(k)}(C_n)} \sum_{t=1}^n E\left[(\boldsymbol{s_t} - \boldsymbol{\theta}_{1:n}[t])^2\right] = \Omega\left(\min\{n, n^{\frac{1}{2k+3}}C_n^{\frac{2}{2k+3}}\}\right),$$

*where the expectation is taken wrt to randomness in the strategy of the player and $\epsilon_t$.*

We define linear forecasters to be strategies that predict a fixed linear function of the history. This includes a large family of polices including the ARIMA family, Exponential Smoothers for Time Series forecasting, Restarting OGD etc. However in the presence of spatially inhomogeneous

smoothness – which is the case with TV bounded sequences – these policies are doomed to perform sub-optimally. This can be made precise by providing a lower-bound on the minimax regret for linear forecasters. Since the offline problem of smoothing is easier than that of forecasting, a lower-bound on the minimax MSE of linear smoother will directly imply a lower-bound on the regret of linear forecasting strategies. By the results of [Donoho et al., 1998], we have the following proposition:

**Proposition 2** (Minimax regret for linear forecasters). *Linear forecasters will suffer a dynamic regret of at least $\Omega(n^{1/(2k+2)})$ for forecasting sequences that belong to $TV^k(1)$.*

Thus we must look in the space of policies that are *non-linear* functions of past labels to achieve a minimax dynamic regret that can potentially match the lower-bound in Proposition 1.

### 4.2 Policy

In this section, we present our policy and capture the intuition behind its design. First, we introduce the following notations.

- The policy works by partitioning the time horizon into several bins. $t_h$ denotes start time of the current bin and $t$ be the current time point.
- $\boldsymbol{W}$ denotes the orthonormal Discrete Wavelet Transform (DWT) matrix obtained from a CDJV wavelet construction [Cohen et al., 1993] using wavelets of regularity $k + 1$.
- $T(\boldsymbol{y})$ denotes the vector obtained by elementwise soft-thresholding of $\boldsymbol{y}$ at level $\sigma\sqrt{\beta \log l}$ where $l$ is the length of input vector.
- $\boldsymbol{x_t} \in \mathbb{R}^{(k+1)}$ denotes the vector $[1, t - t_h + k + 1, \ldots, (t - t_h + k + 1)^k]^T$.
- $A_t = \boldsymbol{I} + \sum_{s=t_h-k}^{t} \boldsymbol{x_s}\boldsymbol{x_s}^T$
- $\texttt{recenter}(\boldsymbol{y}[s : e])$ function first computes the Ordinary Least Square (OLS) polynomial fit with features $\boldsymbol{x}_s, \ldots, \boldsymbol{x}_e$. It then outputs the residual vector obtained by subtracting the best polynomial fit from the input vector $\boldsymbol{y}[s : e]$.
- Let $L$ be the length of a vector $\boldsymbol{u}_{1:t}$. $\texttt{pack}(\boldsymbol{u})$ first computes $l = \lfloor \log_2 L \rfloor$. It then returns the pair
  $(\boldsymbol{u}_{1:2^l}, \boldsymbol{u}_{t-2^l+1:t})$. We call elements of this pair as segments of $\boldsymbol{u}$.

---

`Ada-VAW`: inputs - observed $y$ values, TV order $k$, time horizon $n$, sub-gaussian parameter $\sigma$, hyper-parameter $\beta > 24$ and $\delta \in (0, 1]$

1. For $t = 1$ to $k - 1$, predict 0
2. Initialize $t_h = k$
3. For $t = k$ to $n$:
   (a) Predict $\hat{y}_t = \langle \boldsymbol{x_t}, A_t^{-1} \sum_{s=t_h-k}^{t-1} y_s \boldsymbol{x_s} \rangle$
   (b) Observe $y_t$ and suffer loss $(\hat{y}_t - \boldsymbol{\theta}_{1:n}[t])^2$
   (c) Let $\mathbf{y}_r = \texttt{recenter}(\mathbf{y}[t_h - k : t])$ and $L$ be its length
   (d) Let $(\boldsymbol{y}_1, \boldsymbol{y}_2) = \texttt{pack}(\boldsymbol{y}_r)$
   (e) Let $(\hat{\boldsymbol{\alpha}}_1, \hat{\boldsymbol{\alpha}}_2) = (T(\boldsymbol{W}\mathbf{y}_1), T(\boldsymbol{W}\mathbf{y}_2))$
   (f) Restart Rule: If $\|\hat{\boldsymbol{\alpha}}_1\|_2 + \|\hat{\boldsymbol{\alpha}}_2\|_2 > \sigma$ then
      i. set $t_h = t + 1$

---

The basic idea behind the policy is to adaptively detect intervals that have low $TV^k$ distance. If the $TV^k$ distance within an interval is guaranteed to be low enough, then outputting a polynomial fit can suffice to obtain low prediction errors. Here we use the polynomial fit from VAW [Vovk, 2001] forecaster in step 3(a) to make predictions in such low $TV^k$ intervals. Step 3(e) computes denoised wavelets coefficients. It can be shown that the expression on the LHS of the inequality in step 3(f) can be used to lower bound $\sqrt{L}$ times the $TV^k$ distance of the underlying ground truth with high probability. Informally speaking, this is expected as the wavelet coeffcients for a CDJV system with regularity $k$ are computed using higher order differences of the underlying signal. A restart is triggered when the scaled $TV^k$ lower-bound within a bin exceeds the threshold of $\sigma$. Thus we use

the energy of denoised wavelet coefficients as a device to detect low $TV^k$ intervals. In Appendix E we show that popular padding strategies such as zero padding, greatly inflate the $TV^k$ distance of the recentered sequence for $k \geq 1$. This hurts the dynamic regret of our policy. To obviate the necessity to pad for performing the DWT, we employ a packing strategy as described in the policy.

## 4.3 Performance Guarantees

**Theorem 3.** *Consider the the feedback model* $y_t = \boldsymbol{\theta}_{1:n}[t] + \epsilon_t$ $t = 1, \ldots, n$ *where* $\epsilon_t$ *are independent* $\sigma^2$ *subguassian noise and* $|\boldsymbol{\theta}_{1:n}[t]| \leq B$. *If* $\beta = 24 + \frac{8 \log(8/\delta)}{\log(n)}$, *then with probability at least* $1 - \delta$, `Ada-VAW` *achieves a dynamic regret of* $\tilde{O}\left(n^{\frac{1}{2k+3}} \left(n^k \|D^{k+1}\boldsymbol{\theta}_{1:n}\|_1\right)^{\frac{2}{2k+3}}\right)$ *where* $\tilde{O}$ *hides poly-logarithmic factors of* $n$, $1/\delta$ *and constants* $k, \sigma, B$ *that do not depend on* $n$.

*Proof Sketch.* Our proof strategy falls through the following steps.

1. Obtain a high probability bound of bias variance decomposition type on the total squared error incurred by the policy within a bin.

2. Bound the variance by optimally bounding the number of bins spawned.

3. Bound the squared bias using the restart criterion.

Step 1 is achieved by using the subgaussian behaviour of revealed labels $y_t$. For step 2, we first connect the wavelet coefficients of a recentered signal to its $TV^k$ distance using ideas from theory of Regression Splines. Then we invoke the "uniform shrinkage" property of soft thresholding estimator to construct a lowerbound of the $TV^k$ distance within a bin. Such a lowerbound when summed across all bins leads to an upperbound on the number of bins spawned. Finally for step 3, we use a reduction from the squared bias within a bin to the regret of VAW forecaster and exploit the restart criterion and adpative minimaxity of soft thresholding estimator [Donoho et al., 1998] that uses a CDJV wavelet system. $\qquad \square$

**Corollary 4.** *Consider the setup of Theorem 3. For the problem of forecasting sequences* $\boldsymbol{\theta}_{1:n}$ *with* $n^k \|D^{k+1}\boldsymbol{\theta}_{1:n}\|_1 \leq C_n$ *and* $\|\boldsymbol{\theta}_{1:n}\|_\infty \leq B$, `Ada-VAW` *when run with* $\beta = 24 + \frac{8 \log(8/\delta)}{\log(n)}$ *yields a dynamic regret of* $\tilde{O}\left(n^{\frac{1}{2k+3}} (C_n)^{\frac{2}{2k+3}}\right)$ *with probability atleast* $1 - \delta$.

**Remark 5.** *(Adaptive Optimality) By combining with trivial regret bound of* $O(n)$, *we see that dynamic regret of* `Ada-VAW` *matches the lower-bound provided in Proposition 1.* `Ada-VAW` *optimally adapts to the variational budget* $C_n$. *Adaptivity to time horizon* $n$ *can be achieved by the standard doubling trick.*

**Remark 6.** *(Extension to higher dimensions) Let the ground truth* $\boldsymbol{\theta}_{1:n}[t] \in \mathbb{R}^d$ *and let* $\boldsymbol{v}_i = [\boldsymbol{\theta}_{1:n}[1][i], \ldots, \boldsymbol{\theta}_{1:n}[n][i]], \Delta_i = n^k \|D^{k+1}\boldsymbol{v}_i\|_1$ *for each* $i \in [d]$. *Let* $\sum_{i=1}^{d} \Delta_i \leq C_n$. *Then by running* $d$ *instances of* `Ada-VAW` *in parallel where instance* $i$ *predicts ground truth sequence along co-ordinate* $i$, *a regret bound of* $\tilde{O}\left(d^{\frac{2k+1}{2k+3}} n^{\frac{1}{2k+3}} C_n^{\frac{2}{2k+3}}\right)$ *can be achieved.*

**Remark 7.** *(Generalization to other losses) Consider the protocol in Figure 1. Instead of squared error losses in step (5), suppose we use loss functions* $f_t(x)$ *such that* $\arg\min f_t(x) = \boldsymbol{\theta}_{1:n}[t]$ *and* $f'_t(x)$ *is* $\gamma$*-Lipschitz. Under this setting,* `Ada-VAW` *yields a dynamic regret of* $\tilde{O}\left(\gamma n^{\frac{1}{2k+3}} C_n^{\frac{2}{2k+3}}\right)$ *with probability at least* $1 - \delta$. *Concrete examples include (but not limited to):*

1. *Huber loss,* $f_t^{(\omega)}(x) = \begin{cases} 0.5(x - \boldsymbol{\theta}_{[1:n]}[t])^2 & |x - \boldsymbol{\theta}_{[1:n]}[t]| \leq \omega \\ \omega(|x - \boldsymbol{\theta}_{[1:n]}[t]| - \omega/2) & otherwise \end{cases}$ *is 1-Lipschitz in gra-dient.*

2. *Log-Cosh loss,* $f_t(x) = \log(\cosh(x - \boldsymbol{\theta}_{[1:n]}[t]))$ *is 1-Lipschitz in gradient.*

3. $\epsilon$*-insensitive logistic loss [Dekel et al., 2005],* $f_t^{(\epsilon)}(x) = \log(1 + e^{x - \boldsymbol{\theta}_{[1:n]}[t] - \epsilon}) + \log(1 + e^{-x + \boldsymbol{\theta}_{[1:n]}[t] - \epsilon}) - 2\log(1 + e^{-\epsilon})$ *is 1/2-Lipschitz in gradient.*

The rationale behind both Remark 6 and Remark 7 is described at the end of Appendix C.2

**Proposition 8.** *There exist an $O\left(((k+1)n)^2\right)$ run-time implementation of* Ada-VAW.

The run-time of $O(n^2)$ is larger than the $O(n \log n)$ run-time of the more specialized algorithm of [Baby and Wang, 2019] for $k = 0$. This is due to the more complex structure of higher order CDJV wavelets which invalidates their trick that updates the Haar wavelets in an amortized $O(1)$ time.

# 5 Extensions

In this section, we discuss the potential applications of the proposed algorithm which broadens its generalizability to several interesting use cases.

## 5.1 Optimality for Higher Order Sobolev and Holder Classes

So far we have been dealing with total variation classes, which can be thought of as $\ell_1$-norm of the $(k+1)$th order derivatives. An interesting question to ask is "how does Ada-VAW behave under smoothness metric defined in other norms, e.g., $\ell_2$-norm and $\ell_\infty$-norm?" Following [Tibshirani, 2014], we define the higher order discrete Sobolev class $\mathcal{S}^{k+1}(C'_n)$ and discrete Holder class $\mathcal{H}^{k+1}(L'_n)$ as follows.

$$\mathcal{S}^{k+1}(C'_n) = \{\boldsymbol{\theta}_{1:n} : n^k \|D^{k+1}\boldsymbol{\theta}_{1:n}\|_2 \leq C'_n\},$$
$$\mathcal{H}^{k+1}(L'_n) = \{\boldsymbol{\theta}_{1:n} : n^k \|D^{k+1}\boldsymbol{\theta}_{1:n}\|_\infty \leq L'_n\},$$

where $k \geq 0$. These classes feature sequences that are *spatially more regular* in comparison to the higher order $TV^k$ class. It is well known that (see for eg. [Gyorfi et al., 2002]) the following embedding holds true:

$$\mathcal{H}^{k+1}\left(\frac{C_n}{n}\right) \subseteq \mathcal{S}^{k+1}\left(\frac{C_n}{\sqrt{n}}\right) \subseteq TV^k(C_n).$$

Here $\frac{C_n}{\sqrt{n}}$ and $\frac{C_n}{n}$ are respectively the maximal radius of a Sobolev ball and Holder ball enclosed within a $TV^k(C_n)$ ball. Hence we have the following Corollary.

**Corollary 9.** *Assume the observation model of Theorem 3 and that $\boldsymbol{\theta}_{1:n} \in \mathcal{S}^{k+1}(C'_n)$. If $\beta = 24 + \frac{8 \log(8/\delta)}{\log(n)}$, then with probability at least $1 - \delta$,* Ada-VAW *achieves a dynamic regret of $\tilde{O}\left(n^{\frac{2}{2k+3}}[C'_n]^{\frac{2}{2k+3}}\right)$.*

It turns out that this is the optimal rate for the Sobolev classes, even in the easier, offline non-parametric regression setting [Gyorfi et al., 2002]. Since a Holder class can be sandwiched between two Sobolev balls of same minimax rates [see, e.g., Gyorfi et al., 2002], this also implies the adaptive optimality for the Holder class. We emphasize that Ada-VAW does not need to know the $C_n, C'_n$ or $L'_n$ parameters, which implies that it will achieve the smallest error permitted by the right norm that captures the smoothness structure of the unknown sequence $\boldsymbol{\theta}_{1:n}$.

## 5.2 Optimality for the case of Exact Sparsity

Next, we consider the performance of Ada-VAW on sequences satisfying an $\ell_0$-(pseudo)norm measure of the smoothness, defined as

$$\mathcal{E}^{k+1}(J_n) = \{\boldsymbol{\theta}_{1:n} : \|D^{k+1}\boldsymbol{\theta}_{1:n}\|_0 \leq J_n, \|\boldsymbol{\theta}_{1:n}\|_\infty \leq B\}.$$

This class captures sequences that has at most $J_n$ jumps in its $(k+1)^{th}$ order difference, which covers (modulo the boundedness) $k$th order discrete splines [see, e.g., Schumaker, 2007, Chapter 8.5] with exactly $J_n$ knots, and arbitrary piecewise polynomials with $O(J_n/k)$ polynomial pieces.

The techniques we developed in this paper allows us to establish the following performance guarantee for Ada-VAW, when applied to sequences in this family.

**Theorem 10.** *Let $y_t = \boldsymbol{\theta}_{1:n}[t] + \epsilon_t$, for $t = 1, \ldots, n$ where $\epsilon_t$ are iid sub-gaussian with parameter $\sigma^2$ and $\|D^{k+1}\boldsymbol{\theta}_{1:n}\|_0 \leq J_n$ with $|\boldsymbol{\theta}_{1:n}[t]| \leq B$ and $J_n \geq 1$. If $\beta = 24 + \frac{8 \log(8/\delta)}{\log(n)}$, then with probability at least $1 - \delta$,* Ada-VAW *achieves a dynamic regret of $\tilde{O}(J_n)$ where $\tilde{O}$ hides polynmial factors of $\log(n)$ and $\log(1/\delta)$.*

We also establish an information-theoretic lower bound that applies to all algorithms.

**Proposition 11.** *Under the interaction model in Figure 1, the minimax dynamic regret for forecasting sequences in $\mathcal{E}^{k+1}(J_n)$ is $\Omega(J_n)$.*

**Remark 12.** *Theorem 10 and Proposition 11 imply that* `Ada-VAW` *is optimal (up to logarithmic factors) for the sequence family $\mathcal{E}^k(J_n)$. It is noteworthy that the* `Ada-VAW` *is adaptive in $J_n$, so it is essentially performing as well as an oracle that knows* how many *knots are enough to represent the input sequence as a discrete spline and* where *they are in advance (which leaves only the $J_n$ polynomials to be fitted).*

## 6 Conclusion

In this paper, we considered the problem of forecasting $TV^k$ bounded sequences and proposed the first efficient algorithm – `Ada-VAW`– that is adaptively minimax optimal. We also discussed the adaptive optimality of `Ada-VAW` in various parameters and other function classes. In establishing strong connections between the locally adaptive nonparametric regression literature to the adaptive online learning literature in a concrete problem, this paper could serve as a stepping stone for future exchanges of ideas between the research communities, and hopefully spark new theory and practical algorithms.

## Acknowledgment

The research is partially supported by a start-up grant from UCSB CS department, NSF Award #2029626 and generous gifts from Adobe and Amazon Web Services.

## Broader Impact

1. Who may benefit from the research? This work can be applied to the task of estimating trends in time series forecasting. For example, financial firms can use it to do stock market predictions, distribution sector can use it do inventory planning, meterological observatories can use it for weather forecast and health and planning sector can forecast the spread of contagious diseases etc.

2. Who may be put at disadvantage? Not applicable

3. What are the consequences of failure of the system? There is no system to speak off, but failure of the strategy can lead to financial losses for the firms deploying the strategy to do forecasting. Under the assumptions stated in the paper though, the technical results are formally proven and come with the stated mathematical guarantee.

4. Method leverages the biases in data? Not applicable.

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
