[Supplementary Material]

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

# A  Background

In this section, we compile some preliminary results well established in literature. For brevity we only discuss the essential aspects that lead to design of our algorithm and its proof.

## A.1  Non-parametric regression

A popular model studied in non-parametric regression is

$$y_i = f(i/n) + \epsilon_i, i \in [n], \tag{2}$$

where $\epsilon_i$ are iid subgaussian noise and for unknown $f : [0,1] \to \mathbb{R}$. The idea is to recover the underlying ground truth $f$ from the observations $y_i$. Let $\boldsymbol{\theta}_{1:n} = [f(1/n), \ldots, f(1)] \in \mathbb{R}^n$ be the ground truth sequence. We constraint the ground truth to belong to some non-parametric class. A well studied (dating back since 90s atleast) non-parametric family is the class of $TV^k$ bounded sequences defined below.

$$\mathrm{TV}^k(C_n) := \{\boldsymbol{\theta}_{1:n} \in \mathbb{R}^n : n^k \|D^{k+1}\boldsymbol{\theta}_{1:n}\|_1 \leq C_n\}.$$

The sequences in this class have a piecewise (discrete) polynomial structure. Each stable section features a polynomial of degree atmost $k$. However the number of polynomial sections and positions where the sequence transitions from one polynomial to another is unknown. This makes the task of estimating ground truth from noisy observations quite challenging. Moreover as noted in [Kim et al., 2009], such sequences can be used to model a wide spectrum of real world phenomena. As noted in Section 2, such $TV^k$ sequences can be obtained by sampling the function whose continuous $TV^k$ distance is bounded. An illustration for $k = 2$ is given in Figure 3.

The purpose of a non-parametric regression algorithm $\mathcal{A}$ is to estimate $\boldsymbol{\theta}_{1:n}$ given the noisy observations $y_i$. The most common metric used to ascertain the performance of an algorithm in non-parametric regression literature is the squared error loss. Let the estimates of the algorithm be $\hat{\boldsymbol{y}}_{1:n}$. The empirical risk is defined as

$$R_n = E\left[\sum_{t=1}^n (\hat{\boldsymbol{y}}_{1:n}[t] - \boldsymbol{\theta}_{1:n}[t])^2\right],$$

and the minimax risk for estimating sequences in $TV^k(C_n)$ is formulated as

$$R_n^* = \min_{\mathcal{A}} \max_{\boldsymbol{\theta} \in TV^k(C_n)} R_n,$$

where $\mathcal{A}$ is an estimation of algorithm. It is well established (see eg. [Donoho et al., 1998]) that

$$R_n^* = \Omega(n^{\frac{1}{2k+3}} C_n^{\frac{2}{2k+3}}). \tag{3}$$

## A.2  Wavelet Smoothing

Let $\mathbb{Z}_+ = \mathbb{N} \cup \{0\}$ and $L_2[0,1]$ be the space of all square integrable functions defined in $[0,1]$.

**Definition 13.** *A Multi Resolution Analysis (MRA) on interval [0,1] is a sequence of subspaces $\{V_j, j \in \mathbb{Z}_+\}$ satisfying*

1. *$V_j \subset V_{j+1}$*

2. *$f(x) \in V_j$ if and only if $f(2x) \in V_{j+1}$*

3. *$\bigcap_{j \in \mathbb{Z}_+} = \{0\}$ and $\bigcup_{j \in \mathbb{Z}_+}$ spans $L_2[0,1]$.*

4. *There exists a function $\phi \in V_0$ such that $\{\phi(x - k) : k \in \mathbb{Z}$ such that $\phi(x - k)$ is supported in [0,1]\}$ is an orthonormal basis for $V_0$*

Figure 3: *A $TV^2$ bounded sequence $\boldsymbol{\theta}_{1:n}$ can be obtained by sampling the continuous piecewise quadratic function on the left at points $i/n$, $i \in [n]$. On the right, we plot the $TV^2$ distance of the generated sequence for various sequence lengths $n$. As $n$ increases the discrete $TV^2$ distance converges to a constant value given by the continous $TV^2$ distance of the function on left panel.*

The spaces $V_j$ form an increasing sequence of approximations to $L_2[0,1]$. Let $\phi_{jk}(x) = 2^{j/2}\phi(2^j x - k)$. In what follows we define $\phi_{jk}(x) = 0$ if it is not supported entirely within $[0,1]$. Due to properties 2 and 4 it follows that $\{\phi_{jk}(x), k \in \mathbb{Z}\}$ is an orthonormal basis for $V_j$. The function $\phi(x)$ is called the *scale function*.

Now let's define wavelets. *Detail subspace* $W_j \subset L_2[0,1]$ is defined as the orthogonal complement of $V_j$ in $V_{j+1}$. A function $\psi(x)$ is defined to be a *wavelet* (or mother wavelet) function if $\{\psi_{jk}(x) = 2^{j/2}\psi(2^j x - k), k \in \mathbb{Z}_+$ such that $\psi_{jk}(x)$ is supported in $[0,1]\}$ is an orthonormal basis for $W_j$ $\forall j \in \mathbb{Z}_+$.

**Definition 14.** *A wavelet function $\psi(x)$ has regularity $r$ if*

$$\int_0^1 x^p \psi(x) dx = 0, p = 0, \ldots, r-1.$$

The CDJV construction in [Cohen et al., 1993] is an algorithm that provides a scale function $\phi(x)$ and wavelet function $\psi(x)$ of a given regularity $r$. We record an important property of this construction.

**Proposition 15.** *The CDJV construction with regularity $r$ satisfy*

1. *Let $L = \lceil \log 2r \rceil$. Then $V_L$ contains polynomials of degree $\leq r-1$.*

2. *The functions $\psi_{jk}(x), j \geq L, k \in \mathbb{Z}$ are orthogonal to polynomials of degree atmost $r-1$.*

Let $n = 2^J$ and $L < J$. A discrete Wavelet Transform (DWT) matrix $\boldsymbol{W} \in \mathbb{R}^{n \times n}$ is generated by sampling the basis functions that make up $V_L$ and $W_L, \ldots, W_{J-1}$ at points $i/n, i \in [n]$ and scaling them by a factor of $n^{-1/2}$. The obtained matrix $\boldsymbol{W}$ can be shown to be orthonormal. The total number of basis functions that make up the space $V_J$ is $n$.

Now to provide a clearer picture, we orchestrate all the above ideas with the help of the simple *Haar wavelets*.

**Definition 16.** *The Haar MRA on [0,1] is defined by*

1. *The scale function $\phi(x) = 1$*

2. *The mother wavelet $\psi(x) = -1$ if $x \leq 1/2$; $1$ otherwise.*

3. *Both $\phi(x), \psi(x)$ are zero outside $[0,1]$*

Here $V_0$ is the space of constant signals in $[0,1]$. $W_0$ is the functions of the form $c\psi(x)$ for $c \in \mathbb{R}$. $W_1$ is spanned by $\psi_{10}(x)$ and $\psi_{11}(x)$ and so on. It is clear that regularity of Haar wavelet $\psi(x)$ is 1. In fact Haar system is a special case of CDJV construction for regularity 1. Hence $L = \lceil \log 2r \rceil = 1$. The space $V_1$ is spanned by $\{\phi(x), \psi(x)\}$. It is easy to verify that space $V_1$ contains all polynomials

of degree $r - 1 = 0$ as asserted by Proposition 15. Furthermore property 2 stated in Proposition 15 is also true.

Now let's construct the orthonormal Haar DWT matrix $\boldsymbol{W} \in \mathbb{R}^{n \times n}$. Let $J = \log n$ We need to sample sample basis functions of $V_1, W_1, \dots W_{J-1}$ at points $i/n, i \in [n]$ and scale them by $n^{-1/2}$. For simplicity we illustrate this for $n = 4$.

$$
\boldsymbol{W} = \begin{bmatrix} 1/2 & 1/2 & 1/2 & 1/2 \\ 1/2 & 1/2 & -1/2 & -1/2 \\ 1/\sqrt{2} & -1/\sqrt{2} & 0 & 0 \\ 0 & 0 & 1/\sqrt{2} & -1/\sqrt{2} \end{bmatrix}.
$$

It is noteworthy that general CDJV wavelets for regularity $r \geq 2$ do not have a closed form expression like the Haar system. The filter coefficients are computed by an efficient iterative algorithm.

Define the soft thresholding operator as

$$
T_\lambda(x) = \begin{cases} 0 & |x| \leq \lambda \\ x - \lambda & x > \lambda \\ x + \lambda & x < \lambda \end{cases}
$$

If the input is a vector the operation is done co-ordinate wise.

Now we are ready to discuss the famous universal soft thresholding algorithm of [Donoho et al., 1998].

---

WaveletSoftThreshold: Inputs - observations $\boldsymbol{y}_{1:n}$, subgaussian parameter $\sigma$ of noise in (2), TV order $k$

1. Let $\boldsymbol{W} \in \mathbb{R}^{n \times n}$ be a CDJV DWT matrix of regularity $k + 1$.
2. Output $\hat{\boldsymbol{y}}_{1:n} = \boldsymbol{W}^T T_{\sigma\sqrt{2\log n}}(\boldsymbol{W}y)$.

---

We have the following proposition due to [Donoho et al., 1998].

**Proposition 17.** *The risk of the wavelet soft thresholding scheme satisfy*

$$
R_n = \tilde{O}(n^{\frac{1}{2k+3}} C_n^{\frac{2}{2k+3}}).
$$

Comparing with equation (3) we see that WaveletSoftThreshold is a near minimax algorithm for estimating sequences in $TV^k(C_n)$. It optimally adapts to the unknown radius $C_n$ as well.

### A.3 Vovk Azoury Warmuth (VAW) forecaster

The VAW algorithm is shown in Figure 4. For a more elaborate discussion on this algorithm, refer to chapter 11 of [Cesa-Bianchi and Lugosi, 2006]. The VAW forecaster is defined as follows.

---

VAW algorithm

1. Adversary reveals $\boldsymbol{x}_t \in \mathbb{R}^d$.
2. Agent predicts $\hat{p}_t = \hat{\boldsymbol{w}}_{t-1}^T \boldsymbol{x}_t$ with $\hat{\boldsymbol{w}}_t = (\boldsymbol{I} + \sum_{s=1}^t \boldsymbol{x}_s \boldsymbol{x}_s^T)^{-1} \sum_{s=1}^{t-1} y_s \boldsymbol{x}_s$.
3. Adversary reveals $y_t$.
4. Incur loss $(\hat{p}_t - y_t)^2$.

---

Figure 4: The VAW algorithm

We have the following guarantee on the regret bound of VAW.

**Proposition 18.** *If the VAW forecaster is run on a sequence* $(\boldsymbol{x}_1, y_1), \ldots, (\boldsymbol{x}_n, y_n) \in \mathbb{R}^d \times \mathbb{R}$, *then for all* $\boldsymbol{u} \in \mathbb{R}^d$ *and* $n \geq 1$,

$$\sum_{t=1}^{n}(y_t - \hat{p}_t)^2 - (y_t - \boldsymbol{u}^T \boldsymbol{x}_t) \leq \frac{1}{2}\|\boldsymbol{u}\|_2^2 + \frac{dY^2}{2}\log(1 + \frac{nX^2}{d}),$$

*where* $\|\boldsymbol{x}_t\|_2 \leq X$, *and* $|y_t| \leq Y, t \in [n]$.

# B  Detailed Discussion of Related Literature

In this section, we discuss the connections of our work to existing literature. Throughout this paper when we refer as $\tilde{O}(n^{\frac{1}{2k+3}})$ as optimal regret we assume that $C_n = n^k \|D^{k+1}\boldsymbol{\theta}_{1:n}\|_1$ is $O(1)$.

Table 2: *Summary of regret bounds for* Ada-VAW *run with a fixed input parameter* $k$ *alongside bounds for various other policies.* Ada-VAW *is adaptively optimal for an array of distinct sequence classes featuring varying degrees of smoothness. We assume similar assumptions as in the description of Table 1. We adopt the notation* $a \wedge b = \min\{a, b\}$.

| Sequence Class | Dynamic Regret | | |
|---|---|---|---|
| | Ada-VAW | ARROWS | MA/OGD/Ader |
| $TV^k(C_n):$ $n^k\|D^{k+1}\boldsymbol{\theta}_{1:n}\|_1 \leq C_n$ | $\tilde{O}\left(n^{\frac{1}{2k+3}}C_n^{\frac{2}{2k+3}}\right)$ | $\tilde{O}\left(n^{1/3}C_n^{2/3}\right)$ | $\tilde{O}\left(\sqrt{nC_n}\right)$ |
| $\mathcal{S}^{k+1}\left(\frac{C_n}{\sqrt{n}}\right):$ $n^k\|D^{k+1}\boldsymbol{\theta}_{1:n}\|_2 \leq \frac{C_n}{\sqrt{n}}$ | $\tilde{O}\left(n^{\frac{1}{2k+3}}C_n^{\frac{2}{2k+3}}\right)$ | $\tilde{O}\left(n^{1/3}C_n^{2/3}\right)$ | $\tilde{O}\left(n^{1/3}C_n^{2/3}\right)$ |
| $\mathcal{H}^{k+1}\left(\frac{C_n}{n}\right):$ $n^k\|D^{k+1}\boldsymbol{\theta}_{1:n}\|_\infty \leq \frac{C_n}{n}$ | | | |
| $\mathcal{E}^{k+1}(J_n):$ $\|D^{k+1}\boldsymbol{\theta}_{1:n}\|_0 \leq J_n,\ J_n \geq 1$ | $\tilde{O}(J_n)$ | $\tilde{O}\left(n^{1/3}J_n^{2/3}\right)$ | $\tilde{O}\left(\sqrt{nJ_n}\right)$ |

**Non parametric Regression** As noted in Section 1, the problem setup we consider in this paper can be regarded as an online version of the batch non parametric regression framework. It has been established (see for eg, [Mammen and van de Geer, 1997, Donoho et al., 1998, Tibshirani, 2014]) that minimax rate for estimating sequences with bounded $TV^k$ distance under squared error loss scales as $n^{\frac{1}{2k+3}}(n^k\|D^{k+1}\boldsymbol{\theta}_{1:n}\|_1)^{\frac{2}{2k+3}}$ modulo logarithmic factors of $n$. However, the problem of forecasting is more challenging than the offline setup because while making a prediction, we do not see the noisy realizations of ground truth for the future time points. In this work we connect together several ideas from online learning and batch regression setting to achieve a $\tilde{O}\left(n^{\frac{1}{2k+3}}(n^k\|D^{k+1}\boldsymbol{\theta}_{1:n}\|_1)^{\frac{2}{2k+3}}\right)$ minimax dynamic regret for the forecasting problem.

**Non-stationary Stochastic Optimization** As mentioned before in Section 1, our forecasting framework can be considered as a special case of non-stationary stochastic optimization setting studied in [Besbes et al., 2015, Chen et al., 2018b]. A path variational constraint $V_n := \sum_{t=1}^{n-1}\|f_{t+1} - f_t\|_\infty$ is defined in [Besbes et al., 2015]. With squared error losses $f_t(x) = (x - \theta_t)^2$ and the boundedness constraint on ground truth in Assumption (A5), it can be shown that $V_n = O(\|D\boldsymbol{\theta}_{1:n}\|_1)$. Then their proposed algorithm namely, Restarting Online Gradient Descend (OGD) yields a dynamic regret of $O\left(n^{1/2}(\|D\theta_{1:n}\|_1)^{1/2}\right)$ for our problem. Due to Proposition 1, we see that the rate wrt $n$ is suboptimal for TV orders $k \geq 0$. Finally to achieve this rate, restarting OGD requires the knowledge of a tight bound on $\|D\boldsymbol{\theta}_{1:n}\|_1$ ahead of time which may not be practical on all occasions. Similar conclusions can be drawn if we consider the work of [Chen et al., 2018b].

**Prediction of Bounded Variation sequences** Our problem setup is identical to that of [Baby and Wang, 2019] except for the fact that they consider forecasting sequences whose zeroth order Total Variation is bounded. Our work can be considered as a generalization to any TV order $k$. As the value of $k$ increases, the sequence becomes more regular and one expects sharper rates for dynamic regret. However the algorithm of [Baby and Wang, 2019] gives a suboptimal regret of $O(n^{1/3})$ for $k \geq 1$ even when both $\|D\boldsymbol{\theta}_{1:n}\|_1$ and $n^k\|D^{k+1}\boldsymbol{\theta}_{1:n}\|_1$ are $O(1)$.

We enumerate the comprehensive list of differences of this work when compared to [Baby and Wang, 2019] for quick reference.

- We work with a strictly general path varaiational that promotes piecewise polynomial structure in the comparator sequence. The path variational in [Baby and Wang, 2019] promotes piecewise constant structures.

- By exploiting connections to regression splines, we formulate a more general restarting rule than [Baby and Wang, 2019].

- We demonstrate that zero padding (and many other padding approaches) prior to computing wavelet transform as done in [Baby and Wang, 2019] will not preserve the higher order total variation, thus lead to *far sub-optimal* results for the current problem. We then propose a novel packing scheme to alleviate this.

- We exploit the structure of CDJV wavelets and present a significantly more involved analysis to obtain *sharper* dynamic regret guarantees. Haar wavelets that worked in [Baby and Wang, 2019], did not work here.

- We characterise the optimality of our algorithm for the case of exact sparsity as done in Section 5.2 which was not studied in [Baby and Wang, 2019]. Sharper dynamic regret guarantees for higher order discrete Sobolev and Holder classes are also obtained.

- We extend the framework to prediction in higher dimensions (Remark 6). We identify a class of loss functions other than squared error losses in which the dynamic regret guarantees of `Ada-VAW` still holds (Remark 7).

To gain some perspective, we present a way to analyse the dynamic regret of existing strategies for our problem. Recall that due to (1) the comparator sequence can be considered to be the ground truth $\boldsymbol{\theta}_{1:n}$. In the univariate setting, most of the existing dynamic regret bounds depends on the variational measure $\|D\boldsymbol{\theta}_{1:n}\|_1$. If we assume that first $k+1$ values of the sequence $\theta_{1:n}$ are zero, then by applying the inequality $\|D^{j-1}\boldsymbol{\theta}_{1:n}\|_1 \leq n\|D^j\boldsymbol{\theta}_{1:n}\|_1$, starting at $j = k+1$ and proceeding iteratively towards $j = 1$, we get $\|D\boldsymbol{\theta}_{1:n}\|_1 \leq n^k\|D^{k+1}\boldsymbol{\theta}_{1:n}\|_1$. This will enable us to get regret bounds for algorithms whose dynamic regret depends on the quantity $\|D\boldsymbol{\theta}_{1:n}\|_1$. The bounds obtained in this manner is shown in Table 1.

Using similar arguments, it can be shown that $\mathcal{S}^{k+1}(C_n/\sqrt{n}) \subseteq \mathcal{S}^1(C_n/\sqrt{n})$ for bounded sequences. This results in the regret bounds for policies other than `Ada-VAW` as displayed in Table 2 for Sobolev and Holder classes.

**Adaptive Online Learning** Our problem can also be cast as a special case of various dynamic regret minimization frameworks such as [Zinkevich, 2003, Hall and Willett, 2013, Besbes et al., 2015, Chen et al., 2018b, Jadbabaie et al., 2015, Hazan and Seshadhri, 2007, Daniely et al., 2015, Yang et al., 2016, Zhang et al., 2018a,b, Chen et al., 2018a]. To the best of our knowledge, none of the algorithms presented in these works can achieve the optimal dynamic regret of $O(n^{\frac{1}{2k+3}})$.

**Competitive Online Non parametric Regression** [Rakhlin and Sridharan, 2014] considers an online learning framework with squared error losses where a sequence $y_1, \ldots, y_n$ is revealed by an adversary and the agent makes prediction $s_t$ at time $t$ that depends only on the past history. They only require the the sequence $\boldsymbol{y_{1:n}}$ to be coordinatewise bounded and no stochastic relations between ground truth and revealed labels are assumed. They consider a regret defined as,

$$R := E\left[\sum_{t=1}^n (y_t - s_t)^2 - \inf_{f \in \mathcal{F}} \sum_{t=1}^n (y_t - f(x_t))^2\right], \tag{4}$$

for a non parametric function class $\mathcal{F}$. If we consider $\mathcal{F}$ as the function class with bounded $TV^k$ distance, then their regret bounds implies an upperbound on the dynamic regret in (1). This can be seen by setting $f(x_t) = \boldsymbol{\theta}_{1:n}[t]$ for $\boldsymbol{\theta}_{1:n} \in TV^k(C_n)$ and $y_t = \boldsymbol{\theta}_{1:n}[t] + \epsilon_t$ for independent

subgaussian $\epsilon_t$, $t = 1, \ldots, n$. Then,

$$R \geq E\left[\sum_{t=1}^n (y_t - s_t)^2 - \sum_{t=1}^n (y_t - \boldsymbol{\theta}_{1:n}[t])^2\right],$$

$$=_{(a)} \sum_{t=1}^n E[s_t^2] + 2E[y_t \boldsymbol{\theta}_{1:n}[t]] - (\boldsymbol{\theta}_{1:n}[t])^2 - 2E[y_t]E[s_t],$$

$$= E\left[\sum_{t=1}^n (s_t - \boldsymbol{\theta}_{1:n}[t])^2\right],$$

where (a) is follows from the fact that the forecaster's prediction $s_t$ is independent of $y_t$.

The results of [Rakhlin and Sridharan, 2014] on Besov spaces with squared error loss establishes that minimax rate for the online setting for the problem at hand is also same as that of the iid batch setting. They prove that minimax rate for Besov spaces indexed by $B_{p,q}^s$ is $O(n^{1/(2s+1)})$ in the univariate case whenever $s \geq 1/2$. The $TV^k(C_n)$ class is sandwiched between two Besov spaces $B_{1,1}^{k+1}$ and $B_{1,\infty}^{k+1}$ for an appropriate scaling of the radius. Since the two Besov spaces has the same minimax rate, the minimax dynamic regret for forecasting $TV^k(C_n)$ sequences in the online setting is also $O(n^{1/2k+3})$. However, the arguments in [Rakhlin and Sridharan, 2015] are non-constructive. They propose a generic recipe based on relaxations of sequential Rademacher complexity for designing optimal online policies. However, we were unable to come up with a relaxation that can lead to computationally tractable forecasters that has the optimal dependence of $n$ and variational budget $\|D^{k+1}\boldsymbol{\theta}_{1:n}\|_1$ on the regret rate.

[Gaillard and Gerchinovitz, 2015] proposes a chaining algorithm to optimally control (4) when $\mathcal{F}$ is taken to be the class of Holder smooth functions. Consequently, their algorithm yields optimal rates for dynamic regret defined in (1) when $\theta_t$ are samples of a Holder smooth function. Such functions are spatially more regular than those present in a TV ball. In section 6.2, we show that our proposed policy Ada-VAW achieves the optimal dynamic regret for Holder spaces enclosed within a higher order TV ball with faster run time complexity.

Other works that can be cast under the setting described in [Rakhlin and Sridharan, 2014] such as [Kotłowski et al., 2016, Koolen et al., 2015] are all unable to achieve the optimal dynamic regret for the problem at hand.

**Classical Time Series Forecasters** Algorithms such as ARMA [Box and Jenkins, 1970] and Hidden Markov Models [Baum and Petrie, 1966] aims to detect recurrent patterns in a stationary stochastic process. However, we focus on surfacing out the hidden trends in a non-stationary stochastic process. Our work is more closely related to the idea of Trend Smoothing, similar in spirit to that of Hodrick-Prescott filter [Hodrick and Prescott, 1997] and [Kim et al., 2009].

**Exact Sparsity** It is established in [Guntuboyina et al., 2017] that Trend Filtering can achieve a total squared error rate of $\tilde{O}(J_n)$ for $\mathcal{E}^{k+1}(J_n)$ (defined in Section 5.2) in the batch setting. In each of the $J_n$ stable sections, the gradient of the polynomial signal is zero almost $k$ times. With the boundedness assumption this yields a TV0 distance almost $B(k+1)$ within a single section. At the change points the TV0 distance encountered is almost $B$. Summing across all $J_n$ sections yields a total TV0 distance of $O(KJ_n)$. This bound on TV0 distance can be used to derive the rates of $O(n^{1/3}J_n^{2/3})$ for ARROWS [Baby and Wang, 2019] and $O(\sqrt{nJ_n})$ for policies presented in [Besbes et al., 2015, Chen et al., 2018b, Zinkevich, 2003, Zhang et al., 2018a]. (See Table 2)

## C   Analysis

### C.1   Connecting wavelet coefficients and higher order $TV^k$ distance

**Lemma 19.** *Let* $\tilde{\boldsymbol{\theta}}_{1:t} = \mathtt{recenter}(\boldsymbol{\theta}_{1:t})$ *and* $(\boldsymbol{a}, \boldsymbol{b}) = \mathtt{pack}(\tilde{\boldsymbol{\theta}}_{1:t})$. *For an orthonormal DWT matrix* $\boldsymbol{W}$,

$$\frac{\|\boldsymbol{W}\boldsymbol{a}\|_2 + \|\boldsymbol{W}\boldsymbol{b}\|_2}{\sqrt{t}} \lesssim t^k \|D^{k+1}\boldsymbol{\theta}_{1:t}\|_1,$$

*where we have subsumed constants that depend only on $k$.*

*Proof.* Consider the truncated power basis with knots at points $\frac{1}{n}, \frac{2}{n}, \ldots, 1$ defined as follows:

$$g_1(x) = 1, \ g_2(x) = x, \ldots, \ g_k(x) = x^k$$

$$g_{k+1+j}(x) = \left(x - \frac{j}{n}\right)_+^k, \ j = 1, \ldots, n-k-1,$$

$x_+ = \max\{x, 0\}$. Since an $t \times t$ matrix $\boldsymbol{G}$ with entries $g_j(\frac{i}{t})$ at the position $(i, j)$ is invertible, we can write any sequence $\boldsymbol{\theta}_{1:t}$ as

$$\boldsymbol{\theta}_{1:t}[i] = \sum_{j=1}^{t} \beta_j g_j\left(\frac{i}{t}\right),$$

for $i = 1, \ldots, t$. From the above equation we see that,

$$t^k \|D^{k+1}\boldsymbol{\theta}_{1:t}\|_1 = k! \sum_{j=k+2}^{t} |\beta_j| \tag{5}$$

Let $\tilde{\boldsymbol{\theta}}_{1:t} = \texttt{recenter}(\boldsymbol{\theta}_{1:t})$. Let $\tilde{\boldsymbol{g}}_j = \texttt{recenter}(\boldsymbol{g}_j)$ where $\tilde{\boldsymbol{g}}_j$ is the $j^{th}$ column of the matrix $\boldsymbol{G}$. Since $\|\boldsymbol{g}_j\|_\infty \leq 1$ we have $\|\tilde{\boldsymbol{g}}_j\|_\infty = O(1)$ where the hidden constant only depends on $k$.

Thus

$$\|\tilde{\boldsymbol{\theta}}_{1:t}\|_\infty = \left\| \sum_{j=k+2}^{t} \beta_j \tilde{\boldsymbol{g}}_j \right\|_\infty,$$

$$\leq \sup_{k+2 \leq i \leq t} \|\tilde{\boldsymbol{g}}_i\|_\infty \sum_{j=k+2}^{t} |\beta_j|,$$

$$\lesssim t^k \|D^{k+1}\boldsymbol{\theta}_{1:t}\|_1, \tag{6}$$

where the last line follows from (5). We subsume a constant that only depends on $k$. Now using $\|\boldsymbol{x}\|_2 \leq \sqrt{m}\|\boldsymbol{x}\|_\infty$ for $\boldsymbol{x} \in \mathbb{R}^m$, we have

$$\frac{\|\tilde{\boldsymbol{\theta}}_{1:t}\|_2}{\sqrt{t}} \lesssim t^k \|D^{k+1}\boldsymbol{\theta}_{1:t}\|_1.$$

We have thus established a lower-bound on the TV using the energy of the OLS residuals. For a vector $\boldsymbol{z}$ let $(\boldsymbol{x}, \boldsymbol{y}) = \texttt{pack}(\boldsymbol{z})$. We have the following relations,

$$\|\boldsymbol{z}\|_2 \geq \sqrt{\frac{\|\boldsymbol{x}\|_2^2 + \|\boldsymbol{y}\|_2^2}{2}},$$

$$\geq \frac{\|\boldsymbol{x}\|_2 + \|\boldsymbol{y}\|_2}{2},$$

where the last line follows from Jensen's inequality and the concavity of $\sqrt{\cdot}$ function.

$\square$

## C.2   Bounding the Regret

Our proof strategy falls through the following steps.

1. Obtain a high probability bound of bias variance decomposition type on the total squared error incurred by the policy within a bin.

2. Bound the variance by optimally bounding the number of bins spawned.

3. Bound the bias using the restart criterion and adaptive minimaxity of soft-thresholding estimator [Donoho et al., 1998].

**Lemma 20.** *(bias-variance bound))* *Let $E[\hat{y}_t] = p_t$. For any bin $[t_h, t_l]$ with $t_h \geq k$ discovered by the policy, we have with probability atleast $1 - \delta/2$*

$$\sum_{t=t_h}^{t_l} (\hat{y}_t - \boldsymbol{\theta}_{1:n}[t])^2 \leq \sum_{t=t_h}^{\bar{t}_l} 2(p_t - \boldsymbol{\theta}_{1:n}[t])^2 + 4\sigma^2(k+1) \log\left(1 + \frac{n^{2k+3}}{k+1}\right) \log(4n^3/\delta).$$

*Proof.* First let's consider an arbitrary interval $[\underline{l}, \bar{l}]$ such that $\underline{l} \geq k$. We proceed to bound the bias and variance of predictions made by a VAW forecaster. Note that the bin $[\underline{l}, \bar{l}]$ is arbitrary and may not be an interval discovered by the policy. The predictions made by VAW forecaster at time $t \in [\underline{l}, \bar{l}]$ is given by,

$$\hat{y}_t = \langle x_t, \tilde{\boldsymbol{A}}_{\boldsymbol{t}}^{-1} \sum_{s=\underline{l}-k}^{t-1} y_s \boldsymbol{x_s} \rangle,$$

where $\tilde{\boldsymbol{A}}_{\boldsymbol{t}} = \boldsymbol{I} + \sum_{s=\underline{l}-k}^{t} \boldsymbol{x_s} \boldsymbol{x_s}^T$.

Let

$$p_t = E[\hat{y}_t],$$

$$= \langle x_t, \tilde{\boldsymbol{A}}_{\boldsymbol{t}}^{-1} \sum_{s=\underline{l}-k}^{t-1} \boldsymbol{\theta}_{1:n}[s\boldsymbol{x_s}\rangle.$$

For notational convenience, define

$$\boldsymbol{X_t} = [\boldsymbol{x_{\underline{l}-k}}, \ldots, \boldsymbol{x_t}]^T. \tag{7}$$

Let

$$\text{Var}(\hat{y}_t) = \sigma^2 \boldsymbol{x_t}^T \tilde{\boldsymbol{A}}_{\boldsymbol{t}}^{-1} \boldsymbol{X_t}^T \boldsymbol{X_t} \tilde{\boldsymbol{A}}_{\boldsymbol{t}}^{-1} \boldsymbol{x_t},$$

$$\leq \sigma^2 \boldsymbol{x_t}^T \tilde{\boldsymbol{A}}_{\boldsymbol{t}}^{-1} \boldsymbol{x_t},$$

$$= \sigma_t^2$$

where the last line is due to $\boldsymbol{X_t}^T \boldsymbol{X_t} \preccurlyeq \tilde{\boldsymbol{A}}_{\boldsymbol{t}}$, where $\boldsymbol{U} \preccurlyeq \boldsymbol{V}$ means $\boldsymbol{V} - \boldsymbol{U}$ is a Positive Semi Definite matrix.

Define a normalized random variable

$$Z_t = \frac{\hat{y}_t - p_t}{\sigma_t}. \tag{8}$$

Thus $Z_t$ is a sub-gaussian random variable with variance parameter 1. By sub-gaussian tail inequality we have,

$$P\left(|Z_t| \geq \sqrt{2\log(4n^3/\delta)}\right) \leq \delta/2n^3,$$

for some $\delta \in (0, 1]$. Noting that length of a bin is almost $n$, an application of uniform bound yields

$$P\left(\sup_{\underline{l} \leq t \leq l} |Z_t| \geq \sqrt{2\log(4n^3/\delta)}\right) \leq \delta/2n^2.$$

Adding and subtracting a $\boldsymbol{\theta}_{1:n}[t]$ to the numerator of (8), we get that with probability atleast $1 - \delta/2n^2$,

$$|\hat{y}_t - \boldsymbol{\theta}_{1:n}[t]| \leq |p_t - \boldsymbol{\theta}_{1:n}[t]| + \sigma_t \sqrt{2\log(4n^3/\delta)}, \forall t \in [\underline{l}, \bar{l}].$$

Hence the squared error within a bin can be bounded in probability as

$$\sum_{t=\underline{l}}^{\bar{l}} (\hat{y}_t - \boldsymbol{\theta}_{1:n}[t])^2 \leq \sum_{t=\underline{l}}^{\bar{l}} 2(p_t - \boldsymbol{\theta}_{1:n}[t])^2 + 4\sigma_t^2 \log(4n^3/\delta), \tag{9}$$

where we used $(a+b)^2 \leq 2a^2 + 2b^2$.

Let's focus on the second term in (9). By lemma 11.11 of [Cesa-Bianchi and Lugosi, 2006] and by following the arguments of proof of Theorem 11.7 there, we get

$$\sum_{t=\underline{l}}^{\bar{l}} \sigma_t^2 \leq \sigma^2 \sum_{d=1}^{k+1} \log(1 + \lambda_d),$$

where $\lambda_d$ are the eigenvalues of the $(k+1) \times (k+1)$ matrix $\tilde{\boldsymbol{A}}_{\bar{l}} - \boldsymbol{I}$. It is well known that $\tilde{\boldsymbol{A}}_{\bar{l}} - \boldsymbol{I}$ has the same nonzero eigenvalues as the Gram matrix $\boldsymbol{G}$ with entries $G_{i.j} = \boldsymbol{x_i}^T \boldsymbol{x_j}$. Note that $\|\boldsymbol{x_t}\|_2^2 \leq n^{2k+2}, \forall t \in [1, n]$. Since the product $\Pi_{d=1}^{k+1}(1 + \lambda_d)$ is maximised when $\lambda_d = (\underline{l} - \bar{l})n^{2k+2}/(k+1) \leq n^{2k+3}/(k+1)$ we have,

$$\sigma^2 \sum_{d=1}^{k+1} \log(1 + \lambda_d) \leq \sigma^2 (k+1) \log(1 + \frac{n^{2k+3}}{k+1}).$$

Thus with probability atleast $1 - \delta/n^2$

$$\sum_{t=\underline{l}}^{\bar{l}} (\hat{y}_t - \boldsymbol{\theta}_{1:n}[t])^2 \leq \sum_{t=\underline{l}}^{\bar{l}} 2(p_t - \boldsymbol{\theta}_{1:n}[t])^2 + 4\sigma^2(k+1) \log\left(1 + \frac{n^{2k+3}}{k+1}\right) \log(4n^3/\delta).$$

As mentioned earlier, the bin $[\underline{l}, \bar{l}]$ can be arbitrary and may not be discovered by policy. However, we want to analyze the Total Squared Error (TSE) incurred within true bins spawned by the policy. A small caveat here is that observations within such true bins satisfy the restart criteria and can't be regarded as independent random variables. To get rid of this problem, we use a uniform bound argument to bound the TSE incurred in all possible $O(n^2)$ bins. This leads to

$$P\left(\sup_{[\underline{l}, \bar{l}]} \sum_{t=\underline{l}}^{\bar{l}} (\hat{y}_t - \boldsymbol{\theta}_{1:n}[t])^2 - \sum_{t=\underline{l}}^{\bar{l}} 2(p_t - \boldsymbol{\theta}_{1:n}[t])^2 - 4\sigma^2(k+1) \log\left(1 + \frac{n^{2k+3}}{k+1}\right) \log(4n^3/\delta) \geq 0\right) \leq \delta/2.$$

$\square$

**Lemma 21.** *(subgaussian wavelet coefficients) Let $(\boldsymbol{y_1}, \boldsymbol{y_2}) = \texttt{pack}(\texttt{recenter}(\boldsymbol{y}))$ for a vector $\boldsymbol{y}$ of observations of length L. Let $(\boldsymbol{\alpha_1}, \boldsymbol{\alpha_2}) = (\boldsymbol{W}\boldsymbol{y_1}, \boldsymbol{W}\boldsymbol{y_2})$ for an orthonormal DWT matrix $\boldsymbol{W}$. Then both $\boldsymbol{\alpha_1}$ and $\boldsymbol{\alpha_2}$ are marginally subgaussian with parameter $4\sigma^2$.*

*Proof.* From the theory of least squares regression,

$$\texttt{recenter}(\boldsymbol{y}) = \boldsymbol{y} - \boldsymbol{X_L}(\boldsymbol{X_L}^T\boldsymbol{X_L})^{-1}\boldsymbol{X_L}^T\boldsymbol{y},$$

where $\boldsymbol{X_L}$ is defined as in (7). Since $L \geq k+1$, $\boldsymbol{X_L}^T\boldsymbol{X_L}$ can be shown to be invertible. (see for eg. lemma 36)

Without loss of generality, we proceed to characterize the sub-gaussian behaviour of the *first* wavelet coefficient of $\boldsymbol{y_1}$. The extension to other wavelet coefficients is straight forward.

Let $\boldsymbol{u}^T$ be the first row of the wavelet transform matrix $\boldsymbol{W}$ whose dimension is compatible to $\boldsymbol{y_1}$. Let's augment $\boldsymbol{u}^T$ as follows.

$$\tilde{\boldsymbol{u}}^T = [\boldsymbol{u}^T, \boldsymbol{0}^T],$$

such that length of $\tilde{\boldsymbol{u}}$ is L.

We have,

$$\boldsymbol{\alpha_1}[0] = \tilde{\boldsymbol{u}}^T y - \tilde{\boldsymbol{u}}^T \boldsymbol{X_L}(\boldsymbol{X_L}^T \boldsymbol{X_L})^{-1} \boldsymbol{X_L}^T \boldsymbol{y}. \qquad (10)$$

(10) along with noisy feedback implies that $\boldsymbol{\alpha_1}[0]$ is a Lipschitz function of $L$ iid subgaussian random variables. Then by Proposition 2.12 from [Johnstone, 2017], $\boldsymbol{\alpha_1}[0]$ is also subgaussian with variance parameter given by the square of Lipschitz constant $\ell^2$ times $\sigma^2$. Since $\boldsymbol{\alpha_1}[0]$ is a linear function of the iid subgaussians we have,

$$\ell = \|\tilde{\boldsymbol{u}} - \boldsymbol{X_L}(\boldsymbol{X_L}^T \boldsymbol{X_L})^{-1} \boldsymbol{X_L}^T \tilde{\boldsymbol{u}}\|_2,$$
$$\leq \|\tilde{\boldsymbol{u}}\|_2 + \|\boldsymbol{X_L}(\boldsymbol{X_L}^T \boldsymbol{X_L})^{-1} \boldsymbol{X_L}^T \tilde{\boldsymbol{u}}\|_2,$$
$$\leq_{(a)} \|\boldsymbol{u}\|_2 + \|\boldsymbol{X_L}(\boldsymbol{X_L}^T \boldsymbol{X_L})^{-1} \boldsymbol{X_L}^T\|_2 \|\boldsymbol{u}\|_2,$$
$$=_{(b)} 2.$$

In (a) we used $\|\boldsymbol{Ax}\|_2 \leq \|\boldsymbol{A}\|_2 \|\boldsymbol{x}\|_2$ where $\|\boldsymbol{A}\|_2$ is the induced matrix norm and the fact that $\|\tilde{\boldsymbol{u}}\|_2 = \|\boldsymbol{u}\|_2$. In (b) we notice that $\|\boldsymbol{u}\|_2 = 1$ as the DWT matrix $\boldsymbol{W}$ is orthonormal and $\|\boldsymbol{X_L}(\boldsymbol{X_L}^T \boldsymbol{X_L})^{-1} \boldsymbol{X_L}^T\|_2 = 1$ since $\boldsymbol{X_L}(\boldsymbol{X_L}^T \boldsymbol{X_L})^{-1} \boldsymbol{X_L}^T$ is a projection matrix.

Similarly it can be shown that $\boldsymbol{\alpha_2}$ is marginally subgaussian with parameter $4\sigma^2$. $\qquad\square$

**Lemma 22.** *(uniform shrinkage) Assume the setting of lemma 21. Let $(\hat{\boldsymbol{\alpha_1}}, \hat{\boldsymbol{\alpha_2}}) = (T(\boldsymbol{\alpha_1}), T(\boldsymbol{\alpha_2}))$ where $T(\cdot)$ is the soft-thresholding operator with threshold $\sigma\sqrt{\beta \log n}$. Then with probability atleast $1 - 2n^{3-\beta/8}$, $|(\hat{\boldsymbol{\alpha_r}})_i| \leq |E[(\boldsymbol{\alpha_r})_i]|$ for each co-ordinate $i$ and $r = 1, 2$. The expectation is taken wrt to randomness in the observations.*

*Proof.* Consider a fixed bin $[\underline{l}, \bar{l}]$. Due to results of lemma 21 and subgaussian tail inequality,

$$P\left(|(\hat{\boldsymbol{\alpha_r}})_i - E[(\boldsymbol{\alpha_r})_i]| \geq \sigma\sqrt{\beta \log n}\right) \leq 2n^{-\beta/8}.$$

Then arguing in the similar lines as in the proof of lemma 15 of Baby and Wang [2019], the result follows. $\qquad\square$

**Lemma 23.** *(bin control) With probability atleast $1 - 2n^{3-\beta/8}$, the number of bins $M$, spawned by the policy is atmost*
$$\min\left\{n, \max\{1, \tilde{O}(n^{\frac{1}{2k+3}}\|n^k D^{(k+1)}\boldsymbol{\theta}_{1:n}\|_1^{\frac{2}{2k+3}})\}\right\} \text{ where } \tilde{O} \text{ hides factors that depend on wavelet}$$
*function, constants that only depend on TV order $k$ and polynomial factors of $\log n$.*

*Proof.* Let $L_i$ be the length of the $i^{th}$ bin. Let $\hat{\boldsymbol{\alpha}}_{1i}, \hat{\boldsymbol{\alpha}}_{2i}$ be the denoised wavelet coefficient segments of the re-centered observations within a bin $i$ as described in the policy and $\boldsymbol{\theta_i}$ be the ground truth vector in bin $i$.

By the policy's restart rule,

$$\frac{\sigma}{\sqrt{L_i}} \leq \frac{1}{\sqrt{L_i}}\left(\|\hat{\boldsymbol{\alpha}}_{1i}\|_2 + \|\hat{\boldsymbol{\alpha}}_{2i}\|_2\right).$$

Due to the uniform shrinkage property specified in lemma 22, we have with probability atleast $1 - 2n^{3-\beta/8}$

$$\frac{\sigma}{\sqrt{L_i}} \leq \frac{1}{\sqrt{L_i}}\left(\|\boldsymbol{\alpha}_{1i}\|_2 + \|\boldsymbol{\alpha}_{2i}\|_2\right),$$
$$\lesssim_{(a)} 2^k L_i^k \|D^{k+1}\boldsymbol{\theta_i}\|_1,$$

where (a) follows due to lemma 19. The factor of $2^k$ is due to the fact that length of vectors $\boldsymbol{\alpha}_{1i}$ or $\boldsymbol{\alpha}_{2i}$ is atmost $2L_i$. The last line implies that when the $TV^k$ distance is zero, Ada-VAW doesn't restart with high probability making $M = 1$.

Rearranging and summing across all bins yields

$$\sum_{i=1}^{M} \frac{\sigma}{L_i^{k+1/2}} \lesssim \|D^{k+1}\boldsymbol{\theta}_{1:n}[t]\|_1.$$

Now applying Jensen's inequality for the convex function $f(x) = \frac{1}{x^{k+1/2}}, x > 0$, we get

$$\sigma M^{\frac{2k+3}{2}} n^{\frac{-(2k+1)}{2}} \lesssim \|D^{k+1}\boldsymbol{\theta}_{1:n}\|_1,$$

where $\lesssim$ subsumes constants that depend only on wavelet functions, TV order $k$ and polynomial factors of $\log n$.

Rearranging the last expression yields the lemma. □

**Lemma 24.** *(Vovk-Azoury-Warmuth regret) If the Vovk-Azoury-Warmuth forecaster with output denoted by $\hat{v}_j$ at time $j$, is run on a sequence*
$(\boldsymbol{w_1}, v_1), \ldots, (\boldsymbol{w_n}, v_n) \in \mathbb{R}^{k+1} \times \mathbb{R}$, *then for all $\boldsymbol{u} \in \mathbb{R}^{k+1}$,*

$$\sum_{j=1}^{t} (\hat{v}_j - v_j)^2 - (\boldsymbol{u}^T \boldsymbol{w_j} - v_j)^2 \leq \frac{1}{2}\|\boldsymbol{u}\|_2^2 + \frac{(k+1)B^2}{2}\log\left(1 + \frac{t^{k+2}}{k+1}\right),$$

$$= \tilde{O}(B^2),$$

*where $B = \max_{i=1,\ldots,t}|y_i|$ and $\boldsymbol{w_j} = [1, j, \ldots, j^k]^T$.*

*Proof.* The first inequality is due to Theorem 11.8 of [Cesa-Bianchi and Lugosi, 2006]. The second equality follows because under the given choice of monomial features, it is shown in Corollary 40 that when $\boldsymbol{u}$ is the coefficient vector of OLS fit, $\|\boldsymbol{u}\|_2^2 = O(B^2)$. □

Next we characterize the optimality of soft-thresholding estimator on $TV^k$ class. The key to this is the Theorem 19 from [Baby and Wang, 2019].

**Theorem 25.** *[Baby and Wang, 2019] Consider the observation model $\breve{\boldsymbol{y}} = \breve{\boldsymbol{\alpha}} + \boldsymbol{Z}$, where $\breve{\boldsymbol{y}} \in \mathbb{R}^n$, $\boldsymbol{Z}$ is marginally subgaussian with parameter $\sigma^2$ and $\breve{\boldsymbol{\alpha}} \in \boldsymbol{A}$ for some solid and orthosymmetric $\boldsymbol{A}$. Let $\hat{\boldsymbol{\alpha}}_\delta$ be the soft thresholding estimator with input $\breve{\boldsymbol{y}}$ and threshold $\delta$. When $\delta = \sigma\sqrt{\beta \log n}$, with probability atleast $1 - 2n^{1-\beta/2}$ the estimator $\hat{\boldsymbol{\alpha}}_\delta$ satisfies*

$$\|\hat{\boldsymbol{\alpha}}_\delta - \alpha\|^2 \leq 8.88\beta(1 + \log(n)) \inf_{\hat{\alpha}} \sup_{\alpha \in A} E[\|\hat{\alpha} - \alpha\|^2].$$

We are interested in the case where $\boldsymbol{A}$ is the space of wavelet coefficients for $TV^k$ bounded fucntions. Since $TV^k$ class is sandwiched between two Besov spaces, it can be shown that $\boldsymbol{A}$ is solid and orthosymmetric (see for eg. [Johnstone, 2017], section 4.8). Note that subtracting a polynomial of degree $k$ has no effect on the $TV^k$ distance. It has been established in lemma 21 that OLS residual are subgaussian with parameter $4\sigma^2$. Hence we are under the observation model of Theorem 25. By the results of [Donoho et al., 1998], we have $\inf_{\hat{\alpha}} \sup_{\alpha \in A} E[\|\hat{\alpha} - \alpha\|^2] = \tilde{O}(n^{\frac{1}{2k+3}}(n^k D\boldsymbol{\theta}_{1:n}\|_1)^{\frac{2}{2k+3}} \sigma^{\frac{4k+4}{2k+3}})$. This along with using a uniform bound across all $O(n^2)$ bins leads to the following Corollary.

**Corollary 26.** *Under the observation model and notations in Theorem 25 but with a subgassuan parameter $4\sigma^2$ when $\boldsymbol{A}$ is the wavelet coefficients of re-centered ground truth within a bin discovered by the policy, then with probability atleast $1 - 2n^{3-\beta/8}$*

$$\|\hat{\boldsymbol{\alpha}}_\delta - \alpha\|^2 = \tilde{O}(n^{\frac{1}{2k+3}}(n^k D\boldsymbol{\theta}_{1:n}\|_1)^{\frac{2}{2k+3}} \sigma^{\frac{4k+4}{2k+3}}).$$

**Lemma 27.** *(bias control) Let $E[\hat{y}_t] = p_t$. For any bin $[t_h, t_l]$, $L = t_l - t_h$, with $t_h \geq k$ discovered by the policy, we have with probability atleast $1 - 2n^{3-\beta/8}$*

$$\sum_{t=t_h}^{\bar{t}_l - 1} (p_t - \boldsymbol{\theta}_{1:n}[t])^2 = \tilde{O}(1) + \tilde{O}\left(L^{\frac{2k+1}{2k+3}}\|D^{k+1}\boldsymbol{\theta}_{t_h-k:t_l-1}\|_1^{\frac{2}{2k+3}}\right) + (p_{t_l} - \boldsymbol{\theta}_{1:n}[t_l])^2.$$

*Proof.* For a bin $[t_h, t_l]$ let

$$T = \sum_{t=t_h}^{t_l} (p_t - \boldsymbol{\theta}_{1:n}[t])^2.$$

Note that $T$ is the squared error incurred by the VAW forecaster when run with the sequence $\boldsymbol{\theta}_{t_h:t_l}$. Let $\boldsymbol{u}$ be the coefficient of the OLS fit using monomial features for the ground truth $[\boldsymbol{\theta}_{t_h-k:t_l-1}]$. Further let's recall/adopt the following notations:

1. $(\boldsymbol{g_1}, \boldsymbol{g_2}) = \texttt{pack}\,(\texttt{recenter}(\boldsymbol{\theta}_{t_h-k:t_l-1}))$;

2. $(\boldsymbol{\alpha_1}, \boldsymbol{\alpha_2}) = (\boldsymbol{W}\boldsymbol{g_1}, \boldsymbol{W}\boldsymbol{g_2})$;

1. [Baby and Wang, 2019] $(\boldsymbol{y_1}, \boldsymbol{y_2}) = \texttt{pack}\,\big(\texttt{recenter}(\boldsymbol{y}_{t_h-k:t_l-1})\big)$;

4. $L = t_l - t_h + k$;

5. $(\hat{\boldsymbol{\alpha_1}}, \hat{\boldsymbol{\alpha_2}}) = (T(\boldsymbol{W}\boldsymbol{y_1}), T(\boldsymbol{W}\boldsymbol{y_2}))$ where $T(\cdot)$ is soft-thresholding operator at threshold $\sigma\sqrt{\beta \log n}$.

$$
\begin{aligned}
T - (p_{t_l} - \boldsymbol{\theta}_{1:n}[t_l])^2 &\leq_{(a)} \sum_{j=t_h-k}^{t_l-1} (\boldsymbol{u}^T \boldsymbol{x_j} - \boldsymbol{\theta}_{1:n}[j])^2 + \tilde{O}(B^2), \\
&\leq_{(b)} \|\boldsymbol{\alpha_1}\|_2^2 + \|\boldsymbol{\alpha_2}\|_2^2 + \tilde{O}(B^2), \\
&\leq_{(c)} \|\hat{\boldsymbol{\alpha_1}}\|_2^2 + \|\hat{\boldsymbol{\alpha_2}}\|_2^2 + \|\hat{\boldsymbol{\alpha_1}} - \boldsymbol{\alpha_1}\|_2^2 + \|\hat{\boldsymbol{\alpha_2}} - \boldsymbol{\alpha_2}\|_2^2 + \tilde{O}(B^2), \\
&\leq_{(d)} \|\hat{\boldsymbol{\alpha_1}}\|_2^2 + \|\hat{\boldsymbol{\alpha_2}}\|_2^2 + \tilde{O}\left( L^{\frac{2k+1}{2k+3}} \|D^{k+1}\boldsymbol{\theta}_{t_h-k:t_l-1}\|_1^{\frac{2}{2k+3}} \sigma^{\frac{4k+4}{2k+3}} \right) + \tilde{O}(B^2), \\[2mm]
&\leq_{(e)} \frac{\sigma^2}{L} + \tilde{O}\left( L^{\frac{2k+1}{2k+3}} \|D^{k+1}\boldsymbol{\theta}_{t_h-k:t_l-1}\|_1^{\frac{2}{2k+3}} \sigma^{\frac{4k+4}{2k+3}} \right) + \tilde{O}(B^2), \\
&= \tilde{O}(1) + \tilde{O}\left( L^{\frac{2k+1}{2k+3}} \|D^{k+1}\boldsymbol{\theta}_{t_h-k:t_l-1}\|_1^{\frac{2}{2k+3}} \right),
\end{aligned}
$$

with probability atleast $1 - 2n^{3-\beta/8}$. Inequality (a) is due to lemma 24, (b) is due to orthonormality of wavelet transform matrix $\boldsymbol{W}$, (c) by triangle inequality, (d) by Corollary 26 and (e) is due to the fact that restart condition is not satisfied in the interior of a bin. $\qquad\square$

**Theorem 3.** *Consider the the feedback model $y_t = \boldsymbol{\theta}_{1:n}[t] + \epsilon_t$ $t = 1, \ldots, n$ where $\epsilon_t$ are independent $\sigma^2$ subguassian noise and $|\boldsymbol{\theta}_{1:n}[t]| \leq B$. If $\beta = 24 + \frac{8\log(8/\delta)}{\log(n)}$, then with probability at least $1 - \delta$,* `Ada-VAW` *achieves a dynamic regret of $\tilde{O}\left( n^{\frac{1}{2k+3}} \left( n^k \|D^{k+1}\boldsymbol{\theta}_{1:n}\|_1 \right)^{\frac{2}{2k+3}} \right)$ where $\tilde{O}$ hides polylogarithmic factors of $n$, $1/\delta$ and constants $k, \sigma, B$ that do not depend on $n$.*

*Proof.* Let $L_i$ be the length of the $i^{th}$ bin $[t_h^{(i)}, t_l^{(i)}]$ discovered by the policy. Let

$$T_i = \sum_{t=t_h^{(i)}}^{t_l^{(i)}} (p_t - \boldsymbol{\theta}_{1:n}[t])^2.$$

From lemma 27 we have with with probability atleast $1 - 2n^{3-\beta/8}$,

$$
\begin{aligned}
T_i &= \tilde{O}(1) + \tilde{O}\left( L_i^{\frac{2k+1}{2k+3}} \|D^{k+1}\boldsymbol{\theta}_{t_h^{(i)}-k:t_l^{(i)}-1}\|_1^{\frac{2}{2k+3}} \right) + (p_{t_l^{(i)}} - \boldsymbol{\theta}_{1:n}[t_l^{(i)}])^2 \\
&= \tilde{O}(1) + \tilde{O}\left( L_i^{\frac{2k+1}{2k+3}} \|D^{k+1}\boldsymbol{\theta}_{t_h^{(i)}-k:t_l^{(i)}-1}\|_1^{\frac{2}{2k+3}} \right),
\end{aligned}
$$

where in the last line we used the fact that ground truths are bounded by $B$.

Now summing the squared bias across all $M$ bins discovered by the policy yields

$$T = \sum_{i=1}^{M} T_i,$$

$$=_{(a)} O(\tilde{M}) + \sum_{i=1}^{M} \tilde{O}\left( L_i^{\frac{2k+1}{2k+3}} \|D^{k+1}\boldsymbol{\theta}_{t_h^{(i)}-k:t_l^{(i)}-1}\|_1^{\frac{2}{2k+3}} \right),$$

$$=_{(b)} \tilde{O}\left( n^{\frac{1}{2k+3}} \|n^k D^{(k+1)}\boldsymbol{\theta}_{1:n}\|_1^{\frac{2}{2k+3}} \right) + \sum_{i=1}^{M} \tilde{O}\left( L_i^{\frac{2k+1}{2k+3}} \|D^{k+1}\boldsymbol{\theta}_{t_h^{(i)}-k:t_l^{(i)}-1}\|_1^{\frac{2}{2k+3}} \right),$$

$$=_{(c)} \tilde{O}\left( n^{\frac{1}{2k+3}} \|n^k D^{(k+1)}\boldsymbol{\theta}_{1:n}\|_1^{\frac{2}{2k+3}} \right) + \tilde{O}\left( \left(\sum_{i=1}^{M} L_i\right)^{\frac{2k+1}{2k+3}} \cdot \left(\sum_{i=1}^{M}\|D^{k+1}\boldsymbol{\theta}_{t_h^{(i)}-k:t_l^{(i)}-1}\|_1\right)^{\frac{2}{2k+3}} \right),$$

$$= \tilde{O}\left( n^{\frac{1}{2k+3}} \|n^k D^{(k+1)}\boldsymbol{\theta}_{1:n}\|_1^{\frac{2}{2k+3}} \right) + \tilde{O}\left( n^{\frac{1}{2k+3}} \|n^k D^{(k+1)}\boldsymbol{\theta}_{1:n}\|_1^{\frac{2}{2k+3}} \right), \tag{11}$$

with probability atleast $1 - 4n^{3-\beta/8}$. Line (a) holds with probability atleast $1 - 2n^{3-\beta/8}$. For (b) we used lemma 23 and it holds with probability atleast $\left(1 - 2n^{3-\beta/8}\right)^2 \geq 1 - 4n^{3-\beta/8}$. For (c) we used Holder's inequality $\boldsymbol{x}^T\boldsymbol{y} \leq \|\boldsymbol{x}\|_p\|\boldsymbol{y}\|_q$ with $p = \frac{2k+3}{2k+1}$ and $q = \frac{2k+3}{2}$.

Since the variance within a bin is $\tilde{O}(\sigma^2)$ as indicated by lemma 20, when summed across all bins we get a total variance of $\tilde{O}(\sigma^2 M)$ which is $\tilde{O}\left( n^{\frac{1}{2k+3}} \|n^k D^{(k+1)}\boldsymbol{\theta}_{1:n}\|_1^{\frac{2}{2k+3}} \right)$ by lemma 23.

A trivial upperbound for $T$ is

$$T \leq n(B^2 + \sigma^2),$$
$$= O(n). \tag{12}$$

Combining (11) (12) and the variance summed across all terms yields

$$T = \tilde{O}\left( \max\left\{ n, n^{\frac{1}{2k+3}} \|n^k D^{(k+1)}\boldsymbol{\theta}_{1:n}\|_1^{\frac{2}{2k+3}} \right\} \right),$$

with probability atleast $1 - 4n^{3-\beta/8} - \delta/2$ where the dependence of $\delta$ in the failure probability is due to that fact that bias variance decomposition in lemma 20 holds with probability atleast $1 - \delta/2$. By setting $\beta = 24 + \frac{8\log(8/\delta)}{\log(n)}$, we get the Theorem 3. □

**Remark 28.** *(Specialization to $k = 0$) When specialized to the case $k = 0$, we recover the optimal rate established in [Baby and Wang, 2019] for the bounded ground truth setting upto constants $B$ and $\sigma$. When $k = 0$, our policy predicts $\frac{y_{t_h}+\ldots+y_{t-1}}{t-t_h+2}$ at time $t$. This is similar to online averaging except that the denominator is now $t - t_h + 2$ instead of $t - t_h$. [Baby and Wang, 2019] also considers the scenario where the point-wise bound on ground truth can increase in time as $O(C_n)$. As hinted by the similarity of Ada-VAW with that of [Baby and Wang, 2019] for $k = 0$ along with the fact that our restart rule also lower-bounds the Total Variation of ground truth with high probability, it is possible to get a regret bound of $\tilde{O}(n^{1/3}C_n^{2/3} + C_n^2)$ for Ada-VAW in this stronger setting.*

**Proposition 1** (Minimax Regret). *Let $y_t = \boldsymbol{\theta}_{1:n}[t] + \epsilon_t$ for $t = 1, \ldots, n$ where $\theta_{1:n} \in TV^{(k)}(C_n)$, $|\boldsymbol{\theta}_{1:n}[t]| \leq B$ and $\epsilon_t$ are iid $\sigma^2$ subgaussian random variables. Let $\mathcal{A}_F$ be the class of all forecasting strategies whose prediction at time $t$ only depends on $y_1, \ldots, y_{t-1}$. Let $\boldsymbol{s_t}$ denote the prediction at time $t$ for a strategy $\boldsymbol{s} \in \mathcal{A}_F$. Then,*

$$\inf_{\boldsymbol{s}\in\mathcal{A}_F} \sup_{\boldsymbol{\theta}_{1:n}\in TV^{(k)}(C_n)} \sum_{t=1}^{n} E\left[(\boldsymbol{s_t} - \boldsymbol{\theta}_{1:n}[t])^2\right] = \Omega\left( \min\{n, n^{\frac{1}{2k+3}} C_n^{\frac{2}{2k+3}}\} \right),$$

*where the expectation is taken wrt to randomness in the strategy of the player and $\epsilon_t$.*

*Proof.* Since a batch non-parametric regression algorithm is allowed to see the entire observations ahead of time, lower bound in the batch setting directly translates to lower bound for $R_{dynamic}$. Let $\mathcal{A}_B$ be the set of all offline regression algorithms. The minimax rates of estimation of $TV^k$ bounded sequences under squared error losses from [Donoho et al., 1998] gives,

$$\inf_{s \in \mathcal{A}_B} \sup_{\boldsymbol{\theta}_{1:n} \in TV^{(k)}(C_n)} \sum_{t=1}^{M} E\left[(\boldsymbol{s_t} - \boldsymbol{\theta}_{1:n}[t])^2\right]$$
$$= \Omega\left(n^{\frac{1}{2k+3}} C_n^{\frac{2}{2k+3}}\right).$$

$\square$

From [Donoho et al., 1990], minimax rates of estimation under squared error losses of sequences that satisfy $|\boldsymbol{\theta}_i| \leq B$ scales as $\min\{nB^2, n\sigma^2\}$. Combining the two bounds yields Proposition 1.

**Proposition 8.** *There exist an $O\left(((k+1)n)^2\right)$ run-time implementation of* Ada-VAW.

*Proof.* Let's describe the computational requirement at each time step. As outlined in Section 11.8 of [Cesa-Bianchi and Lugosi, 2006], we can use Sherman-Morrison formula to compute $A_t^{-1}$ in $O((k+1)^2)$ time. Using the same logic we can compute $(\boldsymbol{X_t}^T \boldsymbol{X_t})^{-1}$ needed by recenter operation incrementally in $O((k+1)^2)$ time. Re-centering operation and computation of wavelet coefficients requires $O(n)$ time per round. Since there are $n$ rounds, the total run-time complexity becomes $O((k+1)^2 n^2)$. $\square$

**Extension to higher dimensions** Consider a variational measure and the setup described in Remark 6. Let $\hat{y}_t^{(i)}$ be the prediction of instance $i$ of Ada-VAW at time $t$. For each $i \in [d]$, we've

$$\sum_{t=1}^{n} (\hat{y}_t^{(i)} - \boldsymbol{\theta}_{1:n}[t][i])^2 = \tilde{O}\left(n^{\frac{1}{2k+3}} \Delta_i^{\frac{2}{2k+3}}\right),$$

by Theorem 3. Summing across all dimensions yields,

$$R_n = \sum_{i=1}^{d} \tilde{O}\left(n^{\frac{1}{2k+3}} \Delta_i^{\frac{2}{2k+3}}\right)$$
$$= \tilde{O}\left(d^{\frac{2k+1}{2k+3}} n^{\frac{1}{2k+3}} C_n^{\frac{2}{2k+3}}\right),$$

where the last inequality follows from applying Holder's inequality $\boldsymbol{x}^T \boldsymbol{y} \leq \|\boldsymbol{x}\|_p \|\boldsymbol{y}\|_q$ to $\sum_{i=1}^{d} 1^{\frac{2k+1}{2k+3}} \Delta_i^{\frac{2}{2k+3}}$ with norms $p = \frac{2k+3}{2k+1}$ and $q = \frac{2k+3}{2}$.

**Extension to general losses** Assume the interaction model in Figure 1. Instead of squared error losses, let the losses be $f_t$ as discussed in Remark 7. Since $f_t$ is gamma smooth, we have

$$f_t(b) \leq f_t(a) + f_t'(a)(b-a) + \frac{\gamma}{2}(b-a)^2. \tag{13}$$

Let $\hat{y}_t$ be the prediction of Ada-VAW at time t and $\boldsymbol{\theta}_t := \theta_{1:n}[t]$. Then regret with this loss function is

$$\sum_{t=1}^{n} f_t(\hat{y}_t) - f_t(\theta_t) \leq \sum_{t=1}^{n} \frac{\gamma}{2}(\hat{y}_t - \theta_t)^2,$$

by (13) and using the fact $f_t'(\theta_t) = 0$. Now the statement in Remark 7 is immediate by appealing to Theorem 3.

## C.3 Exact sparsity

We start by the observation that an exact sparsity (i.e sparsity in the $\|\cdot\|_0$ sense) in the number of jumps of $\|D^{k+1}\boldsymbol{\theta}_{1:n}\|_0$ translates to an exact sparsity in the wavelet coefficients. This is made precise by the following lemma.

**Lemma 29.** *Consider a sequence with $\|D^{k+1}\boldsymbol{\theta}_{1:n}\|_0 = J$. Then both the signals $\boldsymbol{\theta}_{1:n}$ and $\tilde{\boldsymbol{\theta}}_{1:n} = $ `recenter`$(\boldsymbol{\theta}_{1:n})$ can be represented using $O(k + J \log n)$ wavelet coefficients of a CDJV system of regularity $k + 1$.*

*Proof.* Throughout this proof when we say jumps, we refer to jumps in $\|D^{k+1}\boldsymbol{\theta}_{1:n}\|_0$. Let $L = 2^{\lceil \log_2(k+1) \rceil}$. Consider splitting the coefficients $\boldsymbol{\alpha}$ of the DWT transform into two parts: $\boldsymbol{\alpha}_{1:L}$ and $\boldsymbol{\alpha}_{L+1:n}$. By CDJV construction, the wavelets corresponding to indices $L+1, \dots, n$ are all orthogonal to polynomials to degree atmost $k$. The space of polynomials of degree atmost $k$ is contained in the span of wavelets identified by the indices $1, \dots, L$. Though the span of the first $L$ wavelets can also generate other waveforms which are not polynomials as well.

Notice that between two jumps, the underlying signal is a polynomial of degree atmost $k$. By orthogonality property discussed above, wavelet coefficients from the group $\boldsymbol{\alpha}_{L+1:n}$ assume the value zero if the support of corresponding wavelet is a region where the signal behaves as a polynomial. Since there are $J$ jump points and each point is covered by $\log n$ wavelets by the Multi Resolution property, there can be atmost $O(J \log n)$ non zero coefficients from the group $\boldsymbol{\alpha}_{L+1:n}$.

When we subtract the best polynomial fit due to the re-centering operation, it is only going to affect the first $L$ coefficients and keep the remaining unchanged. Hence the re-centered signal can have atmost $O(k + J \log n)$ nonzero coefficients.

$\square$

Due to lemmas 19 and 22, the expression in the LHS of restart rule of the policy lower-bounds the $TV^k$ distance within a bin with high probability. So if a bin lies entirely between two jumps, we do not restart with high probability as the $TV^k$ distance is zero. This lead to the following Corollary.

**Corollary 30.** *Let $y_t = \boldsymbol{\theta}_t + \epsilon_t$, for $t = 1, \dots, n$ where $\epsilon_t$ are sub-gaussian with parameter $\sigma^2$ and $\|D^{k+1}\boldsymbol{\theta}_{1:n}\|_0 = J$ with $|\boldsymbol{\theta}_t| \leq B$. Then with probability at-least $1 - 2n^{3-\beta/8}$ `Ada-VAW` restarts $O(J)$ times.*

In the next Theorem, we characterize the optimality of soft-thresholding estimator in the exact sparsity case.

**Theorem 31.** *Under the setup of Corollary 30, the soft thresholding estimator whose estimates denoted by $\hat{\boldsymbol{\alpha}}_{1:n}$ with threshold set to $\sigma\sqrt{\log n}$ satisfy,*

$$\|\hat{\boldsymbol{\alpha}}_{1:n} - \boldsymbol{\theta}_{1:n}\|_2^2 = \tilde{O}(J\sigma^2),$$

*with probability atleast $1 - 2n^{1-\beta/2}$ where $\tilde{O}$ hides logarithmic factors of $n$.*

*Proof.* Let $\boldsymbol{\alpha}$ denote the DWT coefficients of $\boldsymbol{\theta}_{1:n}$. By Gaussian tail inequality and union bound we have $P(\sup_t |\epsilon_t| \geq \sigma\sqrt{\log n}) \leq 2n^{1-\beta/2}$. Conditioning on the event $\sup_t |\epsilon_t| \leq \sigma\sqrt{\log n}$ we are under the observation model in lemma 17 of Baby and Wang [2019]. Following the results there, with probability atleast $1 - 2n^{1-\beta/2}$ we have,

$$\|\hat{\boldsymbol{\alpha}}_{1:n} - \boldsymbol{\theta}_{1:n}\|_2^2 = \sum_{i=1}^{n} \min\left\{\boldsymbol{\alpha}[i]^2, 16\sigma^2 \log n\right\},$$
$$= \tilde{O}(J\sigma^2),$$

where the last line follows from lemma 29 and the fact that $O(k + J \log n) = O(KJ \log n) = O(J \log n)$.

$\square$

Now using a uniform bound argument across all $O(n^2)$ bins yields the following Corollary.

**Corollary 32.** *Under the observation model and notations in Corollary 30 but with a subgassuan parameter $4\sigma^2$ when $\boldsymbol{\theta}_{1:n}$ is the re-centered ground truth within a bin discovered by the policy, then with probability atleast $1 - 2n^{3-\beta/8}$*

$$\|\hat{\alpha}_\delta - \alpha\|^2 = \tilde{O}(J\sigma^2).$$

With Corollaries 30 and 32, the proof of Theorem 3 can be readily adapted to give Theorem 10.

**Proposition 11.** *Under the interaction model in Figure 1, the minimax dynamic regret for forecasting sequences in $\mathcal{E}^{k+1}(J_n)$ is $\Omega(J_n)$.*

*Proof.* Let $U\{a, b, c\}$ denote a uniform sample from set $\{a, b, c\}$. Consider a ground truth sequence as follows:

1. For t=1, $\boldsymbol{\theta}_1 = U\{-B, 0, B\}$

2. For t = 2 to $J_n + 1$:

    - if $\boldsymbol{\theta}_{t-1} = -B$, $\boldsymbol{\theta}_t = U\{0, B\}$
    - if $\boldsymbol{\theta}_{t-1} = 0$, $\boldsymbol{\theta}_t = U\{-B, B\}$
    - if $\boldsymbol{\theta}_{t-1} = B$, $\boldsymbol{\theta}_t = U\{-B, 0\}$

3. For $t > J_n + 1$, output $\boldsymbol{\theta}_t = \boldsymbol{\theta}_{t-1}$

Such a signal will have $\|D^{k+1}\boldsymbol{\theta}_{1:n}\|_0 \leq J_n$. Let's assume that we reveal this sequence generating process to the learner. Then the Bayes optimal algorithm will suffer a regret of $\Omega(J_n)$. □

**Extension to higher dimensions** Let the ground truth $\boldsymbol{\theta}_{1:n}[t] \in \mathbb{R}^d$ and let $\boldsymbol{v}_i = [\boldsymbol{\theta}_{1:n}[1][i], \ldots, \boldsymbol{\theta}_{1:n}[n][i]], \|D^{k+1}\boldsymbol{v}_i\|_1 \leq J_n, \forall i \in [d]$. Then run $d$ instances of Ada-VAW where instance $i$ is dedicated to track the sequence $v_i$. By appealing to Theorem 10 for each co-ordinate and summing across all $d$ dimensions yields a regret bound of $\tilde{O}(dJ_n)$.

## D    Adapting to lower orders of k

Though the theory of offline non parametric regression with squared error loss is well developed for the complete spectrum of function classes $TV^k(C_n)$ with $k \geq 0$, most of the practical interest is often limited to lower orders of $k$ namely $k = 0, 1, 2, 3$ (see for eg. [Kim et al., 2009, Tibshirani, 2014]). This motivates us to design policies that can perform optimally for these lower TV orders without requiring the knowledge of $k$ beforehand.

Let $\mathcal{E}$ be the event that $|\epsilon_t| \leq \sigma\sqrt{2\log(2n^2)}$ for all $t = 1, \ldots, n$ where $\epsilon_t$ are as presented in Figure 1. By using subgaussian tail inequality and a union bound across all time points, it can be shown that the event $\mathcal{E}$ happens with probability atleast $1 - \frac{1}{n}$.

The basic idea to achieve adaptivity to $k$ is as follows:

> **Meta-Policy:**
> - Instantiate Ada-VAW for $k = 0, 1, 2, 3$ and run them in parallel.
> - Forecast according to an Exponentially Weighted Averages (EWA) ([Cesa-Bianchi and Lugosi, 2006]) over the predictions made by each of the instances. Set the parameter $\eta$ of EWA to $1/4(B + \sqrt{2\log(2n^2)})^2$.

We condition on the event $\mathcal{E}$. The arguments in the proof of Theorem 3 still goes through even if we condition on $\mathcal{E}$. Let the dynamic regret of Ada-VAW for a particular value of $k$ be the random

variable $R_n^{(k)}$. The maximum possible value of $R_n^{(k)}$ is $\kappa n$ for some constant $\kappa$. We have,

$$\mathbb{E}[R_n^{(k)}|\mathcal{E}] = \int_{-\infty}^{\kappa n} r d\mathbb{P}(r),$$

$$\leq \gamma n^{\frac{1}{2k+3}} C_n^{\frac{2}{2k+3}} + \int_{\gamma n^{\frac{1}{2k+3}} C_n^{\frac{2}{2k+3}}}^{\kappa n} r d\mathbb{P}(r),$$

$$\leq \gamma n^{\frac{1}{2k+3}} C_n^{\frac{2}{2k+3}} + \kappa n \cdot \delta,$$

for some constant $\gamma$, where last line follows due to Theorem 3. By choosing $\delta = 1/n$ we get

$$\mathbb{E}[R_n^{(k)}|\mathcal{E}] = \tilde{O}\left(n^{\frac{1}{2k+3}} C_n^{\frac{2}{2k+3}}\right). \tag{14}$$

Let $\hat{y}_t$, be the output of any forecasting strategy at time $t$. Each expert in the meta-policy suffers a loss $(y_t - \hat{y}_t)^2$ for appropriate value of $\hat{y}_t$. Let $\theta_t := \boldsymbol{\theta}_{1:n}[t]$. we have

$$\sum_{t=1}^{n} \mathbb{E}[(y_t - \hat{y}_t)^2|\mathcal{E}] - \mathbb{E}[(y_t - \theta_t)^2|\mathcal{E}], =_{(a)} \sum_{t=1}^{n} \mathbb{E}[(\theta_t - \hat{y}_t)^2|\mathcal{E}] - \mathbb{E}[(\hat{y}_t - \theta_t)^2|\mathcal{E}]\mathcal{E}[\epsilon_t|\mathcal{E}],$$

$$= \sum_{t=1}^{n} \mathbb{E}[(\theta_t - \hat{y}_t)^2|\mathcal{E}], \tag{15}$$

where the last line is simply the expected dynamic regret of the strategy and line (a) is due to independence of $\epsilon_t$ with $\hat{y}_t$.

Let the dynamic regret of the meta-policy be denoted as $R_{meta}$. Since squared error loss $(y_t - \hat{y}_t)^2$ is exponentially concave with parameter $1/4(B + \sqrt{2\log(2n^2)})^2$, Proposition 3.1 of [Cesa-Bianchi and Lugosi, 2006] along with (14) and (15) guarantees that,

$$\mathbb{E}[R_{meta}|\mathcal{E}] = \log 4 + \tilde{O}\left(\min_{k=0,1,2,3} n^{\frac{1}{2k+3}} \left(n^k \|D^{k+1}\theta_{1:n}\|_1\right)^{\frac{2}{2k+3}}\right)$$

Thus we see that expected dynamic regret of the meta-policy adapts to TV order $k$ upto a additive constant of $\log 4$. This additive constant only contributes to a small $O(1/n)$ term if we consider the per round regret.

# E  Problems with padding

In this section, we explain why some commonly used padding schemes can potentially inflate the $TV^k$ distance of the resulting sequence.

## E.1  Zero padding

Consider a sequence $\boldsymbol{\theta}_{1:t}$ such that best polynomial fit of this sequence is uniformly zero. Let $\boldsymbol{\gamma}$ be the zero padded version of $\boldsymbol{\theta}_{1:t}$ such that length of $\boldsymbol{\gamma}$ is a power of 2. Let $\tilde{\boldsymbol{\theta}} = [\boldsymbol{\theta}_{t-k}, \ldots, \boldsymbol{\theta}_t, 0, \ldots, 0]^T \in \mathbb{R}^{2k+2}$. We have,

$$(D^{k+1}\boldsymbol{\gamma})^T = [(D^{k+1}\boldsymbol{\theta}_{1:t})^T, (D^{k+1}\tilde{\boldsymbol{\theta}})^T, 0, 0, \ldots, 0].$$

Due to (6), we have $\|\boldsymbol{\theta}_{1:t}\|_\infty = O(t^k \|D^{k+1}\boldsymbol{\theta}_{1:t}\|_1)$. Hence the existence of $\tilde{\boldsymbol{\theta}}$ term makes $\|D^{k+1}\boldsymbol{\gamma}\|_1 = O(t^k \|D^{k+1}\boldsymbol{\theta}_{1:t}\|_1)$.

## E.2  Mirror image padding

Let $\boldsymbol{\gamma}$ be the mirror image padded version of the re-centered sequence, $\boldsymbol{\theta}_{1:t}$. i.e $\boldsymbol{\gamma} = [\theta_1, \ldots, \theta_t, \theta_t, \theta_{t-1}, \ldots]$ such that its length becomes a power of 2. Then,

$$\|D^{k+1}\boldsymbol{\gamma}\|_1 = 2\|D^{k+1}\boldsymbol{\theta}_{1:t}\|_1 + D^{k+1}[\boldsymbol{\theta}_{t-k}, \ldots, \boldsymbol{\theta}_{t-1}, \boldsymbol{\theta}_t, \boldsymbol{\theta}_t, \boldsymbol{\theta}_{t-1}, \ldots, \boldsymbol{\theta}_{t-k}]^T,$$

$$= 2\|D^{k+1}\boldsymbol{\theta}_{1:t}\|_1 + O(t^k \|D^{k+1}\boldsymbol{\theta}_{1:t}\|_1),$$

where the last line follows from (6).

# F Technical Lemmas

**Lemma 33.** *The procedure* `CalcDetRecurse` *in [Dingle, 2005] is sound.*

*Proof.* We use induction on the dimension of the input square matrix.

**Base case:** when $d = 3$. Assume that $e[0][0]$ is non-zero. Let the matrix be given by

$$\boldsymbol{X} = \begin{bmatrix} e_{00} & e_{01} & e_{02} \\ e_{10} & e_{11} & e_{12} \\ e_{20} & e_{21} & e_{22} \end{bmatrix}$$

The idea is to convert $\boldsymbol{X}$ to an upper triangular matrix. Define:

$$\boldsymbol{Y} = \begin{bmatrix} 1 & \frac{e_{01}}{e_{00}} & \frac{e_{02}}{e_{00}} \\ e_{10} & e_{11} & e_{12} \\ e_{20} & e_{21} & e_{22} \end{bmatrix}$$

So that $\det(\boldsymbol{Y}) = \frac{\det(\boldsymbol{X})}{e_{00}}$. Applying elementary row operations we get

$$\det(\boldsymbol{Y}) = \begin{vmatrix} 1 & \frac{e_{01}}{e_{00}} & \frac{e_{02}}{e_{00}} \\ 0 & e_{11} - e_{10}\frac{e_{01}}{e_{00}} & e_{12} - e_{10}\frac{e_{01}}{e_{00}} \\ 0 & e_{21} - e_{20}\frac{e_{01}}{e_{00}} & e_{22} - -e_{20}\frac{e_{01}}{e_{00}} \end{vmatrix}$$

The inner loop in the procedure `CalcDetRecurse` computes the determinant of the inner $2 \times 2$ sub-matrix by considering the numerator of the fractional terms. Hence the value $v$ return by the recursive call is $\det(\boldsymbol{Y}[1:][1:])e_{00}^2$. So $\det(X) = e_{00}\frac{v}{e_{00}^2} = \frac{v}{e_{00}}$. This is precisely the value returned by the procedure after the final division loop.

When $e_{00}$ is zero, we can swap it with the row whose first element is non-zero and apply the arguments above. If such a swap is not possible, the procedure correctly recognizes the determinant as zero.

**Inductive case:** Assume that procedure is sound for matrices upto dimension $n$. Now define $\boldsymbol{Y}$ as before to set the element $e_{00}$ to one. By similar arguments we obtain that value $v$ returned by the recursive call is $\det(\boldsymbol{Y}[1:][1:])e_{00}^n$. Thus we obtain $\det(\boldsymbol{X}) = \frac{v}{e_{00}^{n-1}}$. This division is performed at the final loop of the procedure.

Here also when $e_{00}$ is zero, the swapping argument similar to the base case can be applied.

$\square$

Consider OLS fit on the inputs $(\boldsymbol{x_1}, y_1), \ldots, \boldsymbol{x_t}, y_t)$ where the features $\boldsymbol{x_j} = [1, j, \ldots, j^m]^T$ and the responses obey $\max_{i=1,\ldots,t}|y_i| = B$. Let the design matrix be

$$\boldsymbol{X}_t = [\boldsymbol{x}_1, \ldots, \boldsymbol{x}_t]^T.$$

**Lemma 34.** $\det(\boldsymbol{X_t}^T \boldsymbol{X_t})$ *is a polynomial in $t$ with degree atmost $(k+1)^2$.*

*Proof.* The procedure `CalcDegreeOfDet` in [Dingle, 2005] can be used to upperbound the degree of determinant. It assumes that while doing the subtractions in procedure `CalcDetRecurse`, the highest degree terms in the corresponding polynomials do not cancel out.

Let $m = k + 1$. Observe that $\boldsymbol{X}_t^T \boldsymbol{X_t}$ can be compactly written as

$$\boldsymbol{X}_t^T \boldsymbol{X_t} = \begin{bmatrix} S_0(t) & S_1(t) & \ldots & S_{m-1}(t) \\ \vdots & \vdots & \ddots & \vdots \\ S_{m-1}(t) & S_m(t) & \ldots & S_{2m-2}(t) \end{bmatrix}, \tag{16}$$

where $S_p(t) = \sum_{n=1}^{t} n^p$.

Let's run procedure `CalcDegreeOfDet` on an $m \times m$ matrix $\boldsymbol{D}$ of degrees arising from $\boldsymbol{X}_t^T \boldsymbol{X}_t$ as below.

$$\boldsymbol{D} = \begin{bmatrix} 1 & 2 & \dots & m \\ \vdots & \vdots & \ddots & \vdots \\ m & m+1 & \dots & 2m-1 \end{bmatrix}$$

Let's define a seed sequence $\{s\}_i$ as the sequence of numbers that can be found the main diagonal of a given matrix, excluding the element at the bottom right corner. The seed sequnece of $\boldsymbol{D}$ is simply $1, 3, \dots, 2m-3$. Let $T_i$ be the element at index $(0,0)$ for the matrix in the $i^{th}$ recursive call. Note that $T_1 = 1$. Tracing the steps through the recursion we get

$$T_2 = s_2 + T_1$$
$$T_3 = s_2 + T_2 + T_1$$
$$\vdots$$
$$T_{m-1} = s_{k-1} + T_{k-2} + \dots + T_1$$

In $m-1$ calls, we will be left with a $2 \times 2$ matrix whose entries are

$$\begin{bmatrix} T_{m-1} & 1 + T_{m-1} \\ 1 + T_{m-1} & 2 + T_{m-1} \end{bmatrix}$$

Now let's start with the winding up procedure. There are $k-3$ wind-ups that need to be performed. Let $u_t$ be the wound up value from the $t^{th}$ winding up step. We have,

$$u_{m-2} = 2 + 2T_{m-1} - T_{m-2}$$
$$u_{m-3} = u_{m-2} - 2T_{m-3}$$
$$u_{m-4} = u_{m-3} - 3T_{m-4}$$
$$\vdots$$
$$u_1 = u_2 - (m-2)T_1$$

Note that $u_1$ is the final output produced by the topmost call to `CalcDegreeOfDet` procedure. These systems can be unrolled to get

$$u_1 = 2 + 2T_{m-1} - (T_{m-2} + 2T_{m-3} + \dots + (m-2)T_1$$
$$= 2 + s_{m-1} + \sum_{i=1}^{m-1} s_i$$

Now using explicit expressions for seed sequence $\{s\}_i$ we get

$$u_1 = 2 + 2m - 3 + (m-1)^2$$
$$= m^2$$
$$= (k+1)^2$$

$\square$

**Lemma 35.** *Let $S_p(t)$ be a polynomial in $t$ defined as $S_p(t) = \sum_{n=1}^t n^p$ where $p$ is a non-negative integer. Then,*

$$(-1)^{p-1} S_p(t-1) = S_p(-t)$$

*Proof.* For $a(t) = \frac{t(t+1)}{2}$, Faulhaber's formula states that

$$\sum_{n=1}^t n^p = \sum_{i=1}^{(p-1)/2} c_i a(t)^{(p+1)/2},$$

when $p$ is odd and

$$\sum_{n=1}^{t} n^p = \frac{t + 0.5}{p+1} \sum_{i=1}^{p/2} (i+1)c_i a(t)^{p/2},$$

when $p$ is even. The the explicit form of $c_i$ can be expressed in terms of Bernoulli numbers.

Note that $a(-t) = a(t-1)$. Substituting this in the formulas yields the lemma. $\qquad\square$

**Lemma 36.** *For a universal constant $H(m)$ that depends only on $m = k + 1$,*

$$\det(\boldsymbol{X}_t^T \boldsymbol{X}_t) = H(m)\, t^m \prod_{i=2}^{m} \left(t^2 - (i-1)^2\right)^{m-i+1}$$

*Proof.* The strategy is to characterize the roots of determinant. For brevity let's denote $\boldsymbol{Z}_t = \boldsymbol{X}_t^T \boldsymbol{X}_t$. Observe that

$$\boldsymbol{Z}_t = \sum_{i=1}^{t} \boldsymbol{x}_i \boldsymbol{x}_i^T, \tag{17}$$

where $x_i = [1, \ldots, i^{m-1}]$. Each update $\boldsymbol{x}_i \boldsymbol{x}_i^T$ increases the rank by atmost 1. After $m$ such updates $\boldsymbol{X}_m$ becomes a square Vandermonde matrix formed by the sequence $\{1, 2, \ldots, m\}$. Since all of the elements in the sequence are distinct $\boldsymbol{X}_m$ is full rank and so is $\boldsymbol{Z}_m$. This implies that each such update $\boldsymbol{x}_i \boldsymbol{x}_i^T$ for $i \leq m$ increased the rank by exactly one.

We can view the equation (17) as a quantity that evolves in time. For $1 \leq i \leq m - 1$, there exists $m - i$ rows in $\boldsymbol{Z}_i$ that are linearly dependent. This means $t = i$ is a root of $\det(\boldsymbol{Z}_t)$ with multiplicity $(m - i)$. By defining $\boldsymbol{x}_0 = [0, \ldots, 0]^T$ for the initial case $t = 0$, all the rows are simply zeroes and multiplicity of the root $t = 0$ is $m$. Thus we have established that $t^m \prod_{i=2}^{m} (t - (i-1))^{m-i+1}$ is a sub-expression of $\det(\boldsymbol{Z}_t)$.

Let's view $\boldsymbol{Z}_t$ as a function of $t$ with $t \in \mathbb{R}$ as displayed in (16). Put $t = -t'$ in (16). Then we have,

$$\boldsymbol{Z}(t') = \begin{bmatrix} S_0(-t') & S_1(-t') & \ldots & S_{m-1}(-t') \\ \vdots & \vdots & \ddots & \vdots \\ S_{m-1}(-t') & S_m(-t') & \ldots & S_{2m-2}(-t') \end{bmatrix}.$$

Hence showing $t' = a$ is a root of $\boldsymbol{Z}(t')$ implies that $t = -a$ is a root of $\boldsymbol{Z}_t$. We have

$$\det(\boldsymbol{Z}(t')) = (-1)^m \begin{vmatrix} -S_0(-t') & -S_1(-t') & \ldots & -S_{m-1}(-t') \\ \vdots & \vdots & \ddots & \vdots \\ -S_{m-1}(-t') & -S_m(-t') & \ldots & -S_{2m-2}(-t') \end{vmatrix}$$

Consider

$$\det(\tilde{\boldsymbol{Z}}(t')) = \begin{vmatrix} -S_0(-t') & -S_1(-t') & \ldots & -S_{m-1}(-t') \\ \vdots & \vdots & \ddots & \vdots \\ -S_{m-1}(-t') & -S_m(-t') & \ldots & -S_{2m-2}(-t') \end{vmatrix}$$

When $t'$ is a non-negative integer, lemma 35 implies that the elements in the matrix above are result of the summation:

$$\sum_{i=0}^{t'-1} (-i)^p = (-1)^p S_p(t' - 1)$$

$$= -S_p(-t'),$$

where we adopt the convention $0^0 = 1$.

Thus we have,

$$\tilde{\boldsymbol{Z}}(t') = \sum_{i=1}^{t'} \boldsymbol{x'}_i \boldsymbol{x'}_i^T,$$

where $\boldsymbol{x'}_i = [1, -(i-1), \ldots, (-(i-1))^{m-1}]$. Let $\boldsymbol{X'}_t = [\boldsymbol{x'}_1, \ldots, \boldsymbol{x'}_t]^T$.

After $m$ updates, we have that $\boldsymbol{X'}_m$ is a square Vandermonde matrix defined by the sequence $\{0, -1, \ldots, -(m-1)\}$. Since each of the elements are distinct, this a full rank matrix and so each update $\boldsymbol{x'}_i \boldsymbol{x'}_i^T$ for $i \leq m$ increased the rank by exactly one leading to $\tilde{\boldsymbol{Z}}(m)$ being full rank.

Using similar arguments as above we see that $t' = i$ is a root of $\det(\tilde{\boldsymbol{Z}}(t'))$ with multiplicity $(m-i)$. This in turn imply that $t = -i$ is a root of $\det(\boldsymbol{Z}_t)$ with multiplicity $(m-i)$. Now we have established that $t^m \prod_{i=2}^{m} \left(t^2 - (i-1)^2\right)^{m-i+1}$ is a sub-expression of $\det(\boldsymbol{Z}_t)$. By lemma 34 we conclude that we have found all roots of the determinant and no further terms depending $t$ can be there.

$\square$

**Remark 37.** *We conjecture that the universal constant $H(m)$ in lemma 36 is the determinant of Hilbert matrix of order $m$.*

**Definition 38.** *Let $\boldsymbol{H}(t)$ be a square matrix with each entry $r_{ij}(t) = \frac{n_{ij}(t)}{d_{ij}(t)}$ for polynomials $n_{ij}(t)$ and $d_{ij}(t)$. We say $r_{ij}(t)$ is* Hilbert-like *if $r_{ij}(t) = O\left(\frac{1}{t^{i+j-1}}\right)$ for all $i, j$.*

**Lemma 39.** *All the elements of $\left(\boldsymbol{X}_t^T \boldsymbol{X}_t\right)^{-1}$ are Hilbert-like when $t \geq m = k + 1$.*

*Proof.* Computation of inverse is essentially a computation of determinants of the matrix and its minors. Each element $(i, j)$ of an inverse matrix is a rational function with numerator being determinant of minor $M_{ij}$ and denominator being the determinant of the original symmetric matrix.

Let $\boldsymbol{Z}_t = \boldsymbol{X}_t^T \boldsymbol{X}_t$ When $t \geq m$ we have from lemma 36 that $\det(\boldsymbol{Z}_t) = \Omega(t^{m^2})$. So it is sufficient to show that $\det(M_{ij}$ is $O(t^{m^2+1-i-j})$. The strategy we follow is same of that in lemma 34.

We follow a 1 based indexing. Since $\boldsymbol{Z}_t$ is symmetric, it is enough to compute the minors when $1 \leq i \leq j \leq m$.

**case 1:** Consider $\det(M_{ij})$ when $1 < i < j < m - 1$. Following the same notations as in the prood of lemma 36, after $m - 2$ calls to `CalDegreeOfDet` we end up with a matrix below.

$$\boldsymbol{F} = \begin{bmatrix} T_{m-2} & 1 + T_{m-2} \\ 1 + T_{m-2} & 2 + T_{m-2} \end{bmatrix} \tag{18}$$

The corresponding seed sequence $\{s\}_i$ is $\{1, 3, 5, \ldots, 2i-3, 2i, 2i+2, \ldots, 2j-2, 2j+1, 2j+3, \ldots, 2m-3\}$. The jumps in the progression is attributed to the deletion of row $i$ and column $j$ for obtaining minor $M_{ij}$.

The final output $u_1$, from the topmost call to `CalDegreeOfDet` is then given by

$$
\begin{aligned}
u_1 &= s_{m-2} + \sum_{i=1}^{m-2} s_i \\
&= 2 + (2m - 3) + (i-1)^2 + (j-i)(j+i-1) + (m+j-1)(m-j-1), \\
&= m^2 + 1 - i - j.
\end{aligned}
$$

So $\det(M_{ij})$ is $O(t^{m^2+1-i-j})$ where the constant in the big-oh only dependents on $m$.

**case 2:** $(1 < i < j = m - 1)$. After $m - 2$ recursion calls we get the matrix below.

$$\boldsymbol{F} = \begin{bmatrix} T_{m-2} & 2 + T_{m-2} \\ 1 + T_{m-2} & 3 + T_{m-2} \end{bmatrix} \tag{19}$$

The seed sequence $\{s\}_i$ is $\{1, 3, \ldots, 2i - 3, 2i, \ldots, 2j - 2\}$. So

$$
\begin{aligned}
u_1 &= 3 + s_{m-2} + \sum_{i=1}^{m-2} s_i, \\
&= 3 + (2m - 4) + (i - 1)^2 + (j - i)(j + i - 1), \\
&= m^2 + 1 - i - j.
\end{aligned}
$$

So $\det(M_{ij})$ is $O(t^{m^2 + 1 - i - j})$.

**case 3:** $(1 < i = j < k - 1)$.

The seed sequence $\{s\}_i$ is $\{1, 3, \ldots, 2i - 3, 2i + 1, \ldots, 2m - 3\}$. At the last step we get a matrix as in equation (18). Hence,

$$
\begin{aligned}
u_1 &= 2 + s_{m-2} + \sum_{i=1}^{m-2} s_i, \\
&= 2 + (2m - 3) + (i - 1)^2 + (m - i - 1)(2i + 1 + m - i - 2), \\
&= m^2 + 1 - i - j.
\end{aligned}
$$

So $\det(M_{ij})$ is $O(t^{m^2 + 1 - i - j})$.

**case 4:** $(i = j = m - 1)$.

The seed sequence $\{s\}_i$ is $\{1, 3, \ldots, 2i - 3\}$. At the last step we get a matrix below.

$$
\boldsymbol{F} = \begin{bmatrix} T_{m-2} & 2 + T_{m-2} \\ 2 + T_{m-2} & 3 + T_{m-2} \end{bmatrix}
$$

So,

$$
\begin{aligned}
u_1 &= 4 + s_{m-2} + \sum_{i=1}^{m-2} s_i, \\
&= 2 + (2i - 3) + (i - 1)^2, \\
&= m^2 + 1 - i - j.
\end{aligned}
$$

So $\det(M_{ij})$ is $O(t^{m^2 + 1 - i - j})$.

**case 5:** $(i = j = m)$.

The seed sequence $\{s\}_i$ is $\{1, 3, \ldots, 2m - 5\}$. At the last step we get a matrix as in equation (18).

$$
\begin{aligned}
u_1 &= 2 + s_{m-2} + \sum_{i=1}^{m-2} s_i, \\
&= 2 + (2m - 5) + (m - 2)^2, \\
&= m^2 + 1 - i - j.
\end{aligned}
$$

So $\det(M_{ij})$ is $O(t^{m^2 + 1 - i - j})$.

**case 6:** $(i = j = 1)$.

The seed sequence $\{s\}_i$ is $\{3, \ldots, 2m - 3\}$. At the last step we get a matrix as in equation (18).

$$u_1 = 2 + s_{m-2} + \sum_{i=1}^{m-2} s_i,$$
$$= 2 + (2m - 3) + (m - 1)^2 - 1,$$
$$= m^2 + 1 - i - j.$$

So $\det(M_{ij})$ is $O(t^{m^2+1-i-j})$.

**case 7:** $(1 < i < k - 1 < j = k)$.

The seed sequence $\{s\}_i$ is $\{1, \ldots, 2i - 3, 2i, \ldots, 2m - 4\}$. At the last step we get a matrix as in equation (18).

So,

$$u_1 = 2 + s_{m-2} + \sum_{i=1}^{m-2} s_i,$$
$$= 2 + (2m - 4) + (i - 1)^2 + (m - i - 1)(2i + k - i - 2),$$
$$= m^2 + 1 - i - j.$$

So $\det(M_{ij})$ is $O(t^{m^2+1-i-j})$.

**case 8:** $(i = 1, j = m)$.

The seed sequence $\{s\}_i$ is $\{2, \ldots, 2m - 4\}$. At the last step we get a matrix as in equation (18).

So,

$$u_1 = 2 + s_{m-2} + \sum_{i=1}^{m-2} s_i,$$
$$= 2 + (2m - 4) + (m - 2)(2 + m - 3),$$
$$= m^2 + 1 - i - j.$$

So $\det(M_{ij})$ is $O(t^{m^2+1-i-j})$.

**case 9:** $(i = 1, j = m - 1)$.

The seed sequence $\{s\}_i$ is $\{2, \ldots, 2m - 4\}$. At the last step we get a matrix as in equation (19).

So,

$$u_1 = 3 + s_{m-2} + \sum_{i=1}^{m-2} s_i,$$
$$= 3 + (2m - 4) + (m - 2)(2 + m - 3),$$
$$= m^2 + 1 - i - j.$$

So $\det(M_{ij})$ is $O(t^{m^2+1-i-j})$.

**case 10:** $(i = 1 < j < m - 1)$.

The seed sequence $\{s\}_i$ is $\{2, \ldots, 2j - 2, 2j + 1, \ldots, 2m - 3\}$. At the last step we get a matrix as in equation (18).

So,

$$u_1 = 2 + s_{m-2} + \sum_{i=1}^{m-2} s_i,$$
$$= 2 + (2m-3) + (j-1)(2+j-2) + (m-j-1)(2j+1+(m-j-2)),$$
$$= m^2 + 1 - i - j.$$

So $\det(M_{ij})$ is $O(t^{m^2+1-i-j})$.

$\square$

With the above lemma, the following Corollary can be readily verified.

**Corollary 40.** *When $\boldsymbol{\theta}_{1:n}$ is such that $\|\boldsymbol{\theta}_{1:n}\|_\infty \leq B = O(1)$, we have $\left\| \left( \boldsymbol{X}_t^T \boldsymbol{X}_t \right)^{-1} \boldsymbol{X}_t^T \boldsymbol{\theta}_{1:n} \right\|_2 = O(1)$.*