[Reviews · NeurIPS 2020]

Review 1

Summary and Contributions: This paper studies the problem of forecasting TV^k bounded sequence and gives an algorithm that has the optimal regret bound. The main contribution is to connect the batch non-parametric regression to online stochastic optimization. Also, the author uses techniques from wavelet computation which is a very interesting connection.

Strengths: The main contribution of this paper is closing the gap between the previous upper and lower bound. Moreover, the techniques and connections developed in the paper are also of independent interest and might be useful for future study in related problems.

Weaknesses: The writing quality of this paper can be improved. Specifically, the introduction can start with a broader picture of the problem and explain more intuitions on the connection of the online estimation and the wavelet techniques etc.

Correctness: The technical content looks valid from the main paper, however, I didn't get a chance to verify all the details in the appendix.

Clarity: The presentation of the paper can be improved. Specifically, the structure of the paper, especially the introduction, is less organized.

Relation to Prior Work: The paper makes a comprehensive comparison with previous works. It would be nice if the author could mention more on the difference between the techniques used in this paper and the previous works.

Reproducibility: Yes

Additional Feedback: After the response phase, considering the additional feedback received, I remain with my initial assessment of the paper.


Review 2

Summary and Contributions: This paper proposes a new method for online trend estimation. The method is designed to operate in a streaming data setting, where it sequentially predicts the elements of a time series, given observations of all previous points. The objective is to minimise the cumulative error in the predictions over some fixed horizon. The authors' proposed method operates in a framework similar to that of Baby and Wang (2019, NeurIPS). In that paper, the time series being predicted should have bounded total variation difference. In the present paper, this restriction is generalised and the authors consider time series with bounds on higher order total variation - i.e. they difference the time series k>1 times and assume a bound on the magnitude of that resulting sequence. In line 246 the authors state this is equivalent to a bound on the l_1 norm of the (k+1)^{th} derivative. The proposed method predicts the next observation as an exponentially weighted function of previous data. Rather than use all of the previously observed data, the algorithm incorporates a wavelet based change detection method to detect substantial non-stationarities and limits how far back in the data to go based on this. This approach is similar in structure to the algorithm in Baby and Wang (2019) but the wavelet based procedure is different. The first difference is in the way that sequence of length 2^p where p is an integer are constructed (necessary for the wavelet transform). The present paper creates two overlapping sequences instead of padding with extra 0s. The second difference is the threshold for declaring a change which is optimised to the more general framework of the present paper. The authors prove an optimal order bound on the accumulated regret of the approach, and discuss its extensions to more complex problems: alternative norm bounds on the (k+1)^{th} derivative, and the setting of sparse changes.

Strengths: The proposed method is an interesting generalisation of the ARROWS algorithm from Baby and Wang (2019). The theory is as far as I can tell correct and guarantees that the authors have a order-optimal approach. The authors present the theoretical contributions nicely alongside those of other work to make their contribution in this regard clear. I am not the most familiar with this area and as such find it a little difficult to comment on the novelty of the contribution - it seems to make material improvements/extensions on the approach of Baby and Wang (2019), and the proof seems to use new ideas related to the particular wavelet transform.

Weaknesses: As I will discuss below, I think the principle weaknesses of the paper are in the clarity of its exposition.

Correctness: As far as I can tell the theoretical results are sound. The intuition behind the method seems appropriate to the problem. There is no empirical work to assess.

Clarity: I think the writing of the paper could be improved, there were several (key) areas I had to read multiple times to get the message. The most important example of this was in the description of the policy in Section 4.2. A good start to making this easier to understand would be a few sentences initially describing the general shape of the algorithm without notation (i.e. constructs estimator and computes residuals -> wavelet decomposition on these residuals -> etc.) When describing the various notation T, x, and functions pack, recenter, if there was more of an explanation as to what they were being used for at this point it would also help the reader. In terms of the notation, I think it would be better to replace $T$ with $T_\beta$ as it was not initially clear to me where the $\beta$ term that controls the high probability bound came in to the algorithm. Further, I think the paper should have a proper introduction paragraph stating its scope and aims and not dive in with a list of notation. Is 'comparator sequence' ever really defined? Section 4.3 giving the performance guarantees just starts with theory may be better structured with some linking text, explaining what the purpose of the theorems is, rather than a sequence of theorem statements and disjoint remarks.

Relation to Prior Work: As the work is so closely related to Baby and Wang (2019) I think a clear paragraph stating "this is what that paper does, and this is how this paper is different" would be really helpful to the reader. It seems to me that the authors have done their due diligence in referencing related work, although I am not an expert in this sub-domain, so other reviewers may be better qualified to comment on this.

Reproducibility: Yes

Additional Feedback: UPDATE: Thank you for your response to our reviewers. In light of the other positive reviews and your commitments to improve the exposition, I have increased my score.


Review 3

Summary and Contributions: This paper considers the non-stationary stochastic optimization with squared error losses and noisy gradient feedback. The main proposal is a new variational constraint that generalizes previous one for dynamic regret analysis. The new contraint captures both the sparsity and intensity of changes in underlying dynamics. From the algorithm side, several seemingly disparate components (VAW forecaster and CDJV wavelet) are combined, which are of interest for online learning and statistical learning communities.

Strengths: + The new variational budget strictly generalizes that of previous studies [Besbes et al., 2015; Baby and Wang, 2019]. It can capture both the sparsity and intensity of changes in underlying dynamics. So in the scenarios that indeed satisfies the piecewise stationary assumptions, the proposed algorithm will enjoy much desired guarantees, both empirically and theoreically. + Several new pieces from the offline statistical community are combined and introduced to the online nonparametric community (to the best of my knowledge), and the analysis is non-trivial and likely useful for other studies. + The justification and the empirical demonstraition on the scaling of n^k are nice, both of them release my concerns on the scaling issue.

Weaknesses: As far as I can see, the results of this paper only hold for the square loss, because the algorithm and analysis heavily rely on the access of an unbiased gradient estimate. The access is due to the special form of the square loss (as shown in line 60). So my question is: can the resulst be generalized to more general function classes? Is there any lower bound justification on the optimality of the results? It seems that the minimax lower bound of Proposition 11 is only for a special case of k=0.

Correctness: Correct as far as I have checked.

Clarity: yes

Relation to Prior Work: yes, it is clearly discussed in the appendix B

Reproducibility: No

Additional Feedback: It would be better to add some high-level proof ideas in Section 4.3, to improve the readbility.

[Author Response · NeurIPS 2020]

1    We thank all the reviewers for the detailed feedback.

2    **R1** We thank the reviewer for appreciating our contributions. We will restructure the introduction as the reviewer
3    suggested.

4    *» Comparison to [3]:* We enumerate the comprehensive list of differences of this work when compared to [3] for quick
5    reference.

- We work with a strictly general path varaiational that promotes piecewise polynomial structure in the comparator sequence. The path variational in [3] promotes piecewise constant structures.
- By exploiting connections to regression splines, we formulate a more general restarting rule than [3].
- We demonstrate that zero padding (and many other padding approaches) prior to computing wavelet transform as done in [3] will not preserve the higher order total variation, thus lead to *far sub-optimal* results for the current problem. We then propose a novel packing scheme to alleviate this.
- We exploit the structure of CDJV wavelets and present a significantly more involved analysis to obtain *sharper* dynamic regret guarantees. Haar wavelets that worked in [3], did not work here.
- We characterise the optimality of our algorithm for the case of exact sparsity as done in section 5.2 which was not studied in [3]. Sharper dynamic regret guarantees for higher order discrete Sobolev and Holder classes are also obtained.
- We extend the framework to prediction in higher dimensions (Remark 6). We identify a class of loss functions other than squared error losses in which the dynamic regret guarantees of `Ada-VAW` still holds (Remark 7). Rationale behind both of these arguments can be found at the end of Appendix C.2.

20    **R2** We thank the reviewer for appreciating the order optimality of our results. We agree that the readers can benefit
21    from style of exposition that the reviewer suggested and we promise to incorporate it in the main paper. Please also see
22    the comment to **R1** for a comparison to [3].

23    *» Comparator sequence:* We note that there are two popular notions of dynamic regret studied in literature as follows.

$$R_{\text{dyn-besbes}}(x_1, \ldots, x_n, f_1, \ldots, f_n) = \sum_{t=1}^{n} f_t(x_t) - f_t(u_t^*),$$

24    where $u_t^*$ is the minimizer of $f_t(x)$ and

$$R_{\text{dyn-zinkevich}}(x_1, \ldots, x_n, f_1, \ldots, f_n, u_1, \ldots, u_n) = \sum_{t=1}^{n} f_t(x_t) - f_t(u_t),$$

25    where $u_1, \ldots, u_n$ is any arbitrary sequence. $R_{\text{dyn-zinkevich}}$ is the object of study in [1] where they consider designing
26    algorithms with dynamic regret guarantees as a function of the path length of the comparator sequence. Of-course with
27    this more general notion of regret, there is no notion of a fixed comparator sequence.

28    In our work, we consider $R_{\text{dyn-besbes}}$ as done in [2]. We note that when $f_t$ are strongly convex, the comparator sequence
29    $u_1^*, \ldots, u_n^*$ is unique and well defined. For our problem, $f_t(x) = (x - \boldsymbol{\theta}_{1:n}[t])^2$ and $\boldsymbol{\theta}_{1:n}$ is $TV^k$ bounded as in
30    Assumption A3.

31    **R3** We thank the reviewer for appreciating the fine technical aspects of our work.

32    *» Other losses:* When the loss functions satisfy the conditions of Remark 7, we can still get dynamic regret guarantees
33    for `Ada-VAW`. However, deriving dynamic regret bounds for more general losses with $TV^k$ bounded comparators (or
34    more generally, comparators that belong to Besov space) is a challenging future work.

35    *» Lower bound:* We would like to mention that Proposition 11 holds for all $k \geq 0$. There is a typo at line 799. It should
36    be $\|D^{k+1}\boldsymbol{\theta}_{1:n}\|_0 \leq J_n$.

37    *» Other:* We will include the high-level proof ideas in Section 4.3 as the reviewer suggested.

38    **References**

39    [1] *Online convex programming and generalized infinitesimal gradient ascent*, Zinkevich, ICML 2003

40    [2] *Non-stationary stochastic optimization*, Besbes et al, In Operations Research 2015

41    [3] *Online Forecasting of Total-Variation-bounded Sequences*, Baby and Wang, NeurIPS 2019


[Meta-Review · NeurIPS 2020]

This paper considers the online trend estimation in the non-stationary stochastic optimization framework, where the comparator sequence satisfy certain variational constraints. The main contribution is a polynomial time policy extending Vovk-Azoury-Warmuth forecaster the achieves the minimax optimal rate for dynamic regret. All reviewers liked the paper, appreciating connecting the batch non-parametric regression to online stochastic optimization, techniques from wavelet computation, a model based on variational constraints which nicely captures sparsity and intensity of changes, and the (asymptotically) optimal algorithm.